# Nested Learning: The Illusion of Deep Learning Architectures

**Ali Behrouz**
Google Research
USA
alibehrouz@google.com

**Meisam Razaviyayn**
Google Research
USA
razaviyayn@google.com

**Peiling Zhong**
Google Research
USA
peilinz@google.com

**Vahab Mirrokni**
Google Research
USA
mirrokni@google.com

## Abstract

Over the last decades, developing more powerful neural architectures and simultaneously designing optimization algorithms to effectively train them have been the core of research efforts to enhance the capability of machine learning models. Despite the recent progresses, particularly in developing Language Models (LMs), there are fundamental challenges and unanswered questions about how such models can *continually learn/memorize, self-improved, and find "effective solutions,"*. In this paper, we present a new learning paradigm, called Nested Learning (NL), that coherently represents a model with a set of nested, multi-level, and/or parallel optimization problems, each of which with its own "*context flow*". NL reveals that existing deep learning methods learns from data through *compressing* their own context flow, and explain how *in-context learning* emerges in large models. NL suggests a path (a new dimension to deep learning) to design more expressive learning algorithms with more "*levels*", resulting in higher-order in-context learning abilities. In addition to its neuroscientifically plausible and mathematically white-box nature, we advocate for its importance by presenting three core contributions: (1) Deep Optimizers: Based on NL, we show that well-known gradient-based optimizers (e.g., Adam, SGD with Momentum, etc.) are in fact associative memory modules that aim to compress the gradients with gradient descent. Building on this insight, we present a set of more expressive optimizers with deep memory and/or more powerful learning rules; (2) Self-Modifying Titans: Taking advantage of NL's insights on learning algorithms, we present a novel sequence model that learns how to modify itself by learning its own update algorithm; and (3) Continuum Memory System: We present a new formulation for memory system that generalizes the traditional viewpoint of "long-term/short-term memory". Combining our self-modifying sequence model with the continuum memory system, we present a learning module, called HOPE, showing promising results in language modeling, continual learning, and long-context reasoning tasks.

## 1 Introduction

This version of the paper has been extensively summarized to fit the page limit of NeurIPS camera ready, and some materials, experiments, better baselines, discussions, and methods are in the full version, which might make some parts of this version hard to follow or cause inconsistencies. To avoid such cases, please read our arXiv version [1].

39th Conference on Neural Information Processing Systems (NeurIPS 2025).

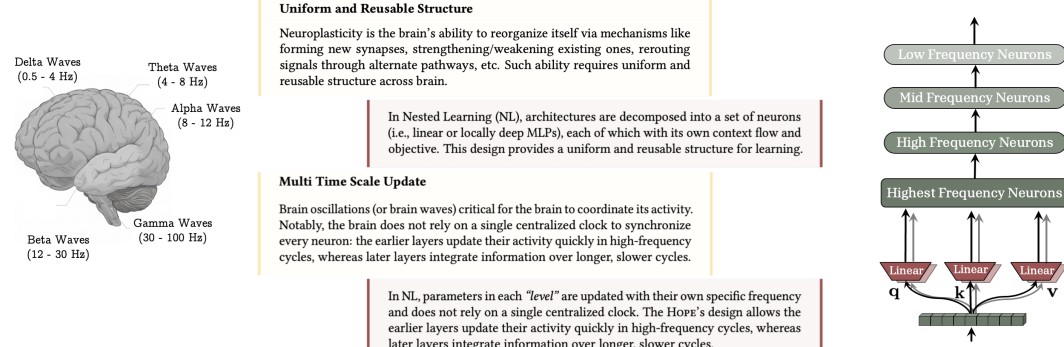

Figure 1: The uniform and reusable structure as well as multi time scale update in the brain are the key components to unlock the continual learning in humans. Nested Learning (NL) allows for multi time-scale update for each component of the brain, while showing that well-known architectures such as Transformers are in fact linear layers with different frequency updates.

For decades, AI research has focused on designing machine learning algorithms that learn from data [2–5] or experience [6–8]; often by optimizing an objective $\mathcal{L}(\boldsymbol{\theta})$ over parameters $\boldsymbol{\theta} \in \Theta$ with gradient-based methods. While traditional machine learning techniques required careful engineering and domain expertise to design feature extractors, limiting their ability to directly process and learn from natural data [9], deep representation learning offered a fully automated alternative to discover the representations needed for the task. Thereafter, deep learning has been an inseparable part of the large-scale computational models with seminal success in chemistry and biology [10], games [11, 12], computer vision [13, 14], and multimodal and natural language understanding [15–17].

Stacking of multiple layers, as it is done in deep learning models, provides the models with larger capacity, better expressive power in representing complex features, and more internal computations (e.g., #FLOPS) [18–20], all of which are critical and desirable characteristics for static tasks that require in-distribution predictions over a previously fixed set. This deep design, however, is not a universal solution to all the challenges and cannot help the expressive power of the models in multiple aspects, for example: (i) The computational depth of deep models might not change with more layers [21, 22], leaving their ability to implement complex algorithms untouched compared to traditional shallow approaches [23]; (ii) The capacity of some class of parameters might show marginal improvement with increasing the depth/width of the model [24]; (iii) The training process might converge to a suboptimal solution, mainly due to the suboptimal choice of the optimizer or its hyperparameters; and (iv) The model's ability to fast adapt to a new task, continually learn, and/or generalize to out-of-distribution data might not changed with stacking more layers and requires more careful designs.

The core part of the efforts to overcome the above challenges and to enhance the capability of deep learning models concentrate on: (1) developing more expressive class of parameters (i.e., neural architectures) [13, 25–28]; (2) introducing objectives that can better model the tasks [29–32]; (3) designing more efficient/effective optimization algorithms to find better solutions or with more resilience to forgetting [33–36]; and (4) scaling the model size to enhance its expressivity, when the "right" choice of architecture, objective, and optimization algorithms are made [24, 37, 38]. Collectively, these advancements and new findings on scaling patterns of deep models have established the foundations upon which Large Language Models (LLMs) have been built.

The development of LLMs marks a pivotal milestone in deep learning research: a paradigm shift from task-specific models to more general-purpose systems with various emergent capabilities as a result of scaling the "right" architectures [38, 39]. Despite all their success and remarkable capabilities in diverse sets of tasks [15, 40, 41], LLMs are largely static after their initial deployment phase, meaning that they successfully perform tasks learned during pre- or post-training, but are unable to continually acquire new capabilities beyond their immediate context. The only adaptable component of LLMs is their *in-context learning* ability–a (known to be emergent) characteristic of LLMs that enables fast adaption to the context and so perform zero- or few-shot tasks [38]. Beyond in-context learning, recent efforts to overcome the static nature of LLMs either are computationally expensive, require external components, lack generalization, and/or might suffer from catastrophic forgetting [42–44], which has led researchers to question if there is a need to revisit how to design machine learning

models and if a new learning paradigm beyond stacking of layers is required to unleash the capabilities of LLMs in continual setups.

**Current Models only Experience the Immediate Present.** As an analogy and to better illustrate the static nature of LLMs, we use the example of anterograde amnesia–a neurological condition where a person cannot form new long-term memories after the onset of the disorder, while existing memories remain intact [45]. This condition limits the person's knowledge and experiences to a short window of present and long past–before the onset of the disorder–which results in continuously experiencing the immediate present as if it were always new. The memory processing system of current LLMs suffer from a similar pattern. Their knowledge is limited to either, the immediate context that fits into their context window, or the knowledge in MLP layers that stores long-past, before the onset of *"end of pre-training."* This analogy, has motivated us to take inspiration from neurophysiology literature and how brain consolidate its short-term memories:

## 1.1 Human Brain Perspective and Neurophysiological Motivation

Human brain is highly efficient and effective when it comes to continual learning (a.k.a. effective context management), which is often attributed to neuroplasticity—the brain's remarkable capacity to change itself in response to new experiences, memories, learning, and even damage [46, 47]. Recent studies support that the formation of Long-term memory involves at least two distinct but complementary consolidation processes [48–50]: (1) A rapid "online" consolidation (also known as synaptic consolidation) phase occurs immediately or soon after learning, even during wakefulness. This is when new and initially fragile memory traces are stabilized and begin transferring from short-term to long-term storage; (2) An "offline" consolidation (also known as systems consolidation) process repeats the replay of the recently encoded patterns—during sharp-wave ripples (SWRs) in the hippocampus, coordinated with cortical sleep spindles and slow oscillations—strengthens and reorganizes the memory and supports transfer to cortical sites [51–53].

Coming back to the analogy of anterograde amnesia, evidence indicates that the condition can impact both stages, but especially the online consolidation phase, mainly due to the fact that hippocampus is the gateway for encoding new declarative memories, and so its damage means new information never will be stored in long-term memory. As mentioned above, the design of LLMs, and more specifically Transformer-based backbones, suffers from a similar condition after the pre-training phase. That is, the information provided in the context, never impacts the long-term memory parameters (e.g., feedforward layers), and so the model is not capable of acquiring new knowledge or skill, unless the information is still stored in the short-term memory (e.g., attention). To this end, although the second stage is equally, or even more, crucial for the consolidation of memories, and its absence can damage the process and might cause loss of memory [54, 55], in this work, we focus on the first stage: memory consolidation as an online process.

**Notations.** We let $x \in \mathbb{R}^{N \times d_{\text{in}}}$ be the input, $\mathcal{M}_t$ represent the state of memory/model $\mathcal{M}$ at time $t$, **K** be the keys, **V** be the values, and **Q** be the query matrices. We use bold lowercase letters with subscript $t$ to refer to the vector corresponds to the input $t$ (i.e., $\boldsymbol{k}_t$, $\boldsymbol{v}_t$, and $\boldsymbol{q}_t$). We further refer to the distribution of any entities $f$ as $p(f)$. Through the paper, we use simple MLPs with $\mathcal{L}_{\mathcal{M}} \geq 1$ layers and residual connection as the architecture of the memory module $\mathcal{M}(\cdot)$. When it is needed, we parameterized the memory module with $\boldsymbol{\theta}_{\mathcal{M}} \supseteq \{W_1, W_2, \ldots, W_{\mathcal{L}_{\mathcal{M}}}\}$, which at least includes the parameters of linear layers in the MLP. We use superscript with parenthesis to refer to parameters in different *levels* of nested learning (different update frequency): i.e., $W^{(\ell)}$.

## 2 Nested Learning

This section discusses the motivations, formal definitions, and general high-level implications of Nested Learning (NL). We start with a formulation of associative memory and then by using step-by-step examples, we build the intuition behind architecture decomposition and its connection to modeling a neural network as an integrated system of optimization problems. We aim to first show how existing methods and concepts in deep learning fall under the NL paradigm and then we present new formulations that go beyond traditional methods and/or provide insights on how to improve existing algorithms and designs.

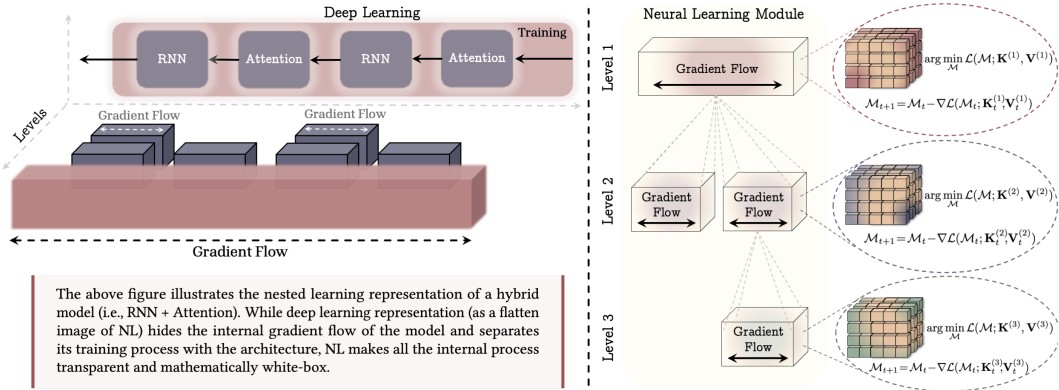

Figure 2: Nested Learning Paradigm that represent a machine learning model and its training procedure as a set of nested optimization problems. (**Left**) An example of Hybrid architecture. While deep learning perspective, as the flattened image of NL, does not provide insight about the depth of computation in the blocks, NL transparently represent all the inner gradient flows. (**Right**) A Neural Learning Module: A computational model that learns how to compress its own context flow. For example, the first level corresponds to the model's most outer-loop training, often refer to as "*pre-training*" step.

## 2.1 Associative Memory

Associative memory—the ability to form and retrieve connections between events—is a fundamental mental process and is an inseparable component of human learning [56]. Often in the literature, the concept of memorization and learning are used interchangeably; in neuropsychology literature, however, these two are clearly distinguished. More specifically, following neuropsychology literature [57], we build our terminology based on the following definition of memory and learning:

> **Learning vs. Memorization:**
>
> *Memory is a neural update caused by an input, and learning is the process for acquiring effective and useful memory.*

In this work, our goal is to first show that all the elements of a computational sequence model, including optimizers and neural networks, are *associative memory systems* that compress their own *context flow*. Broadly speaking, associative memory is an operator that maps a set of keys to a set of values. We follow the general definition of associative memory by Behrouz et al. [58]:

**Definition 1** (Associative Memory). *Given a set of keys $\mathcal{K} \subseteq \mathbb{R}^{d_k}$ and values $\mathcal{V} \subseteq \mathbb{R}^{d_v}$, associative memory is an operator $\mathcal{M} : \mathcal{K} \to \mathcal{V}$ that maps two sets of keys $\mathcal{K}$ and values $\mathcal{V}$. To learn such mapping from the data, an objective $\tilde{\mathcal{L}}(\cdot; \cdot)$ measures the quality of the mapping and $\mathcal{M}$ can be defined as:*

$$\mathcal{M}^* = \arg \min_{\mathcal{M}} \quad \tilde{\mathcal{L}}(\mathcal{M}(\mathcal{K}); \mathcal{V}). \tag{1}$$

While the operator itself is a memory and the mapping acts as a memorization process (i.e., memorizing the connections of events in the context), acquiring such effective operator based on the data, is a learning process. It is notable that, here, keys and values can be any arbitrary events that memory aims to map them and are not limited to tokens. Later in this section, we will discuss that given a context flow, keys and values might be tokens, gradients, sub-sequences, etc. Furthermore, while the term of associative memory is more common in neuroscience and neuropsychology literature, the above formulation is also closely related to data compression and low-dimensional representation. That is, one can interpret the optimization process in Equation 1 as the training process of a network $\mathcal{M}(.)$ that aims to compress the mappings into its parameters and so represent them in a lower dimensional space.

In sequence modeling, where keys and values are input tokens (e.g., tokenized text), the choice of objective and the optimization process for solving Equation 1 can result in distinct sequence

modeling architectures (see [59] and [58]) such as global/local softmax attention [27], or other modern recurrent models [28, 60, 61]. This simple formulation of sequence models provides us with better understanding of their internal process and also a tool to simply compare their modeling power based on their objective and optimization process. In the following, using step-by-step examples, we discuss how this formulation can be applied to all components of a neural architecture (including its optimization process in pre-training) and in fact, how a model is an integrated system of multi-level, nested, and or parallel memories, each of which with its own context flow.

**A Simple Example of MLP Training.** We start with a simple example, in which we aim to train a 1-layer MLP (parameterized with $W$) for task $\mathcal{T}$ and on dataset $\mathcal{D}_{\text{train}} = \{x_1, \ldots, x_{|\mathcal{D}_{\text{train}}|}\}$ by optimizing the objective $\mathcal{L}(\cdot; \cdot)$ with gradient descent. In this case, the training process is equivalent to the following optimization problem:

$$W^* = \arg \min_W \ \mathcal{L}(W; \mathcal{D}_{\text{train}}), \tag{2}$$

whose optimization by gradient descent results in a weight update rule equivalent to:

$$W_{t+1} = W_t - \eta_{t+1} \nabla_{W_t} \mathcal{L}(W_t; x_{t+1}) \tag{3}$$
$$= W_t - \eta_{t+1} \nabla_{y_{t+1}} \mathcal{L}(W_t; x_{t+1}) \otimes x_{t+1}, \quad \text{where } x_{t+1} \sim \mathcal{D}_{\text{train}}, \tag{4}$$

where $y_{t+1} = W x_{t+1}$ is the output of the model for input $x_{t+1}$. Given this formulation, one can let $u_{t+1} = \nabla_{y_{t+1}} \mathcal{L}(W_t; x_{t+1})$ and reformulate the backpropagation process as the solution to an optimization problem on finding an optimal associative memory that maps input data points $\mathcal{D}_{\text{train}} = \{x_t\}_{t=1}^{|\mathcal{D}_{\text{train}}|}$ to their corresponding $u_{t+1} = \nabla_{y_{t+1}} \mathcal{L}(W_t; x_{t+1})$. That is, we let $\mathcal{M}(\cdot) = W_t \cdot$ parametrizes the memory, and use dot-product similarity to measure the quality of $W_t$'s mapping between $x_{t+1}$ and $\nabla_{y_{t+1}} \mathcal{L}(W_t; x_{t+1})$:

$$W_{t+1} = \arg \min_W \ \langle W x_{t+1}, u_{t+1} \rangle + \frac{1}{2\eta_{t+1}} \|W - W_t\|_2^2 \tag{5}$$

$$= \arg \min_W \ \langle W x_t, \nabla_{y_{t+1}} \mathcal{L}(W_t; x_{t+1}) \rangle + \frac{1}{2\eta_{t+1}} \|W - W_t\|_2^2. \tag{6}$$

In the above formulation, $u_{t+1} = \nabla_{y_{t+1}} \mathcal{L}(W_t; x_{t+1})$ can be interpreted as a *local surprise signal in representation space* that quantifies the mismatch between the current output and the structure the objective $\mathcal{L}(\cdot; \cdot)$ enforces. Therefore, this formulation translates the training phase of the model as a process of acquiring effective memory that maps data samples to their Local Surprise Signal (LSS) in representation space–defined as the mismatch between the current output and the structure enforced by the objective $\mathcal{L}(\cdot; \cdot)$. Accordingly, in this example, our model has *a single gradient flow* over the data samples, which is only active over dataset $\mathcal{D}_{\text{train}} = \{x_1, \ldots, x_{|\mathcal{D}_{\text{train}}|}\}$ and will be frozen for any other data samples afterwards (a.k.a inference or test time).

Next, in the above example, we replace the gradient descent algorithm with its enhanced momentum-based variant, resulting in an update rule of:

$$W_{t+1} = W_t - \mathbf{m}_{t+1}, \tag{7}$$
$$\mathbf{m}_{t+1} = \mathbf{m}_t - \eta_{t+1} \nabla_{W_t} \mathcal{L}(W_t; x_{t+1}) = \mathbf{m}_t - \eta_{t+1} \nabla_{y_{t+1}} \mathcal{L}(W_t; x_{t+1}) \otimes x_{t+1}. \tag{8}$$

In Equation 8, given the previous state of Equation 7 (at time $t$), the value of $\nabla_{W_t} \mathcal{L}(W_t; x_{t+1})$ or similarly $\nabla_{y_{t+1}} \mathcal{L}(W_t; x_{t+1})$ are independent of recurrence in Equation 8 and so can be pre-computed beforehand. To this end, we let $u_{t+1} = \nabla_{W_t} \mathcal{L}(W_t; x_{t+1})$, and so Equation 8 can be reformulated as:

$$W_{t+1} = W_t - \mathbf{m}_{t+1}, \tag{9}$$
$$\mathbf{m}_{t+1} = \arg \min_{\mathbf{m}} \ -\langle \mathbf{m}, \nabla_{W_t} \mathcal{L}(W_t; x_{t+1}) \rangle + \eta_{t+1} \|\mathbf{m} - \mathbf{m}_t\|_2^2 \tag{10}$$

$$= \arg \min_{\mathbf{m}} \ -\langle \mathbf{m} x_{t+1}, \nabla_{y_{t+1}} \mathcal{L}(W_t; x_{t+1}) \rangle + \eta_{t+1} \|\mathbf{m} - \mathbf{m}_t\|_2^2, \tag{11}$$

where the optimization problem in Equation 10 is equivalent to on step of gradient descent with adaptive learning rate of $\eta_{t+1}$. Given these formulation, one can interpret the momentum term as either: (1) a key-less associative memory that compress the gradients into its parameters, or (2) an associative memory that learns how to map data points to their corresponding LSS-value. Interestingly, this formulation reveals that gradient descent with momentum is indeed a two-level

optimization process, where the memory is optimized by simple gradient descent algorithm. This process is closely related to Fast Weight Programs (FWPs) [62], where the weight update process (i.e., Equation 9) is the slow network that its momentum weight is generated by a fast network (i.e., Equation 10).

Concluding the above examples, we observed that the training process of a 1-layer MLP with: (1) Gradient descent is a *1-level* associative memory that learns how to map data points to their corresponding LSS-value; and (2) Gradient descent with momentum is a *2-level* associative memory (or optimization process) that the inner-level learns to store gradient values into its parameters, and then the outer-level updates the slow weight (i.e., $W_t$) with the value of the inner-level memory. While these are the most simple examples with respect to both architecture and optimizer algorithms, one might ask if similar conclusion can be made in more complex setups.

**An Example of Architectural Decomposition.** In the next example, we replace the MLP module with a linear attention [60]. That is, we aim to train a 1-layer linear attention for task $\mathcal{T}$ and on a sequence of $\mathcal{D}_{\text{train}} = \{x_1, \ldots, x_{|\mathcal{D}_{\text{train}}|}\}$ by optimizing the objective $\mathcal{L}$ with gradient descent. Recalling the unnormalized linear attention formulation:

$$\boldsymbol{k}_t = x_t W_{\boldsymbol{k}}, \qquad \boldsymbol{v}_t = x_t W_{\boldsymbol{v}}, \qquad \boldsymbol{q}_t = x_t W_{\boldsymbol{q}}, \tag{12}$$

$$\mathcal{M}_t = \mathcal{M}_{t-1} + \boldsymbol{v}_t \boldsymbol{k}_t^\top, \tag{13}$$

$$y_t = \mathcal{M}_t \boldsymbol{q}_t . \tag{14}$$

As discussed in earlier studies [58, 59], the recurrence in Equation 13 can be reformulated as the optimization process of a matrix-valued associative memory $\mathcal{M}_t(\cdot)$, in which, it aims to compress the mappings of keys and values into its parameters. In more details, in Definition 1, if we let $\tilde{\mathcal{L}}(\mathcal{M}_{t-1}; \boldsymbol{k}_t, \boldsymbol{v}_t) := -\langle \mathcal{M}_{t-1} \boldsymbol{k}_t, \boldsymbol{v}_t \rangle$ and aim to optimize the memory with gradient descent, the memory update rule is: (Note that $\nabla \tilde{\mathcal{L}}(\mathcal{M}_{t-1}; \boldsymbol{k}_t, \boldsymbol{v}_t) = \boldsymbol{v}_t \boldsymbol{k}_t^\top$ and we let learning rate $\eta_t = 1$)

$$\mathcal{M}_{t+1} = \arg\min_{\mathcal{M}} \ \langle \mathcal{M} \boldsymbol{k}_{t+1}, \boldsymbol{v}_{t+1} \rangle + \|\mathcal{M} - \mathcal{M}_t\|_2^2 \quad \text{with gradient descent,} \tag{15}$$

$$\Rightarrow \mathcal{M}_{t+1} = \mathcal{M}_t - \nabla \tilde{\mathcal{L}}(\mathcal{M}_t; \boldsymbol{k}_{t+1}, \boldsymbol{v}_{t+1}) = \mathcal{M}_t + \boldsymbol{v}_{t+1} \boldsymbol{k}_{t+1}^\top, \tag{16}$$

which is equivalent to the update rule of an unnormalized linear attention in Equation 13. Also, note that as we observed in the first example, training a linear layer with gradient descent is a 1-layer optimization problem of an associative memory (see Equation 3) and so the general training/updating process of projection layers (i.e., $W_{\boldsymbol{k}}, W_{\boldsymbol{v}}$, and $W_{\boldsymbol{q}}$) is itself an optimization process of associative memory. Accordingly, this setup, i.e., training a linear attention with gradient descent, can be seen as a two-level optimization process, where the outer-loop (also known as training process) optimizes the projection layers with gradient descent, while the inner-loop optimizes the inner memory of $\mathcal{M}_t$ with gradient descent.

Note that, as discussed above, here, we have two associative memories, and so each of which has their own optimization process and gradient flow. That is, in the optimization of outer-level parameters of $W_{\boldsymbol{k}}, W_{\boldsymbol{v}}$, and $W_{\boldsymbol{q}}$ there is no gradient with respect to parameter $\mathcal{M}(\cdot)$ and so there is no backpropagation through it. Similarly, in the inner-level, there is no backpropagation through projection layers and they are considered frozen. Furthermore, it is notable that in this example, the above formulation is also closely connected to FWPs perspective of linear attentions [63], where projections are considered slow weights, and memory update in Equation 13 is the fast weight update rule.

**Architectural Decomposition with More Levels.** In both above examples, we discussed simple cases, where they can be translated into 2-level optimization processes, which also coincides with their FWPs interpretations. In practice, however, we need to use more powerful optimization algorithms to train the model, and/or use more powerful recurrent update rule for memory. As a simple example, assume we use gradient descent with momentum to train a linear attention model. In the above examples, we show that how the linear attention component can be decomposed into two nested optimization problem. Similarly, here the model can be represented as a 2-level optimization problem, where (1) the inner level optimizes the memory to compress the context using gradient descent (see Equation 15), and (2) the outer level optimizes the projection layers with gradient descent with momentum. Interestingly, from the first example, we know that "gradient descent with momentum" algorithm itself is indeed a 2-level optimization problem where the momentum term itself is an associative memory that compress the past gradients into its parameters.

## 2.2 Nested Optimization Problems

In the previous section, we provided examples to demonstrate how one can decompose a machine learning model into a set of nested or multi-level optimization problems. Next, we first aim to present a formal formulation for nested learning problems and then define Neural Learning Module–an integrated computational system that learns from data.

As we observed in the previous section, while we decomposed the model into a set of optimization process, it is still unclear if we can define a hierarchy (or order) over these problems, and uniquely represent the model in this format. Inspired by the hierarchy of brain waves that indicates the information processing frequency rate of each part (discussed in Section 1), we use the update rate of each optimization problem to order the components in multiple levels. To this end, we let the one update step over one data point to be the unit of time, and define the update frequency rate of each component as:

**Definition 2** (Update Frequency). *For any component of A, which can be a parametric component (e.g., learnable weights or momentum term in gradient descent in momentum) or a non-parametric component (e.g., attention block), we define its frequency, denoted as $f_A$, as its number of updates per unit of time.*

Given the above update frequency, we can order the components of a machine learning algorithm based on operator $(\cdot \succ \cdot)$. We let $A$ to be faster than $B$ and denote $A \succ B$ if: **(1)** $f_A > f_B$, or **(2)** $f_A = f_B$ but the computation of the $B$'s state at time $t$ requires the computation of $A$'s state at time $t$. In this definition, when $A \not\succ B$ and $B \not\succ A$, we let $A \overset{f}{=} B$, which indicates that $A$ and $B$ has the same frequency update, but their computation is independent of each other (Later, we provide an example of this cases in AdamW optimizer). Based on the above operator, we sort the components into an ordered set of "*levels*", where (1) components in the same level have the same frequency update, and (2) the higher the level is, the lower its frequency. Notably, based on the above definition, each component has its own optimization problem and so context. While we optimize the component's inner objective with gradient-based optimizers, the above statement is equivalent to having exclusive gradient flow for each component in the model. In general case, however, one can use non-parametric solution (as we later discuss about attention).

**Neural Learning Module.** Given the above definition of nested learning problems, we define neural learning module as a new way of representation of machine learning models that shows the model as an interconnected system of components, each of which with its own gradient flow. Note that, orthogonal to deep learning, nested learning allows us to define neural learning models with more levels, resulting in more expressive architecture.

> *Nested learning allows computational models that are composed of multiple (multi-layer) levels to learn from and process data with different levels of abstraction and time-scales.*

Next, we study optimizers and well-known deep learning architectures from the nested learning perspective, and provide examples that how NL can help to enhance those components.

## 2.3 Optimizers as Learning Modules

In this section, we start by understanding how well-known optimizers and their variants are special instances of nested learning. Recall the gradient descent method with momentum,

$$W_{i+1} = W_i + \mathbf{m}_{i+1}$$
$$\mathbf{m}_{i+1} = \alpha_{i+1}\mathbf{m}_i - \eta_t \nabla \mathcal{L}\left(W_i; x_i\right), \tag{17}$$

where matrix (or vector) $\mathbf{m}_i$ is the momentum at state $i$ and $\alpha_i$ and $\eta_i$ are adaptive learning and momentum rates, respectively. Assuming $\alpha_{i+1} = 1$, the momentum term can be viewed as the result of optimizing the following objective with gradient descent:

$$\min_{\mathbf{m}} \ \langle \mathbf{m} \, \nabla\mathcal{L}(W_i; x_i)^\top, \mathbf{I} \rangle. \tag{18}$$

This interpretation shows that momentum can indeed be viewed as a meta memory module that learns how to memorize gradients of the objective into its parameters. Building on this intuition, we

can show that Adam is the optimal associative memory for the L2 regression objective on models' gradients. Next, we show that how this perspective can result in designing more expressive optimizers:

**Extension: More Expressive Association.** As discussed earlier, momentum is a value-less associative memory and so has limited expressive power. To address this issue, following the original definition of associative memory (i.e., mapping keys to values), we let value parameter $v_i = \mathbf{P}_i$ and so the momentum aims to minimize:

$$\min_{\mathbf{m}} \ \langle \mathbf{m} \, \nabla \mathcal{L}(W_i; x_i)^\top, \mathbf{P}_i \rangle, \tag{19}$$

using gradient descent, resulting in the update rule:

$$W_{i+1} = W_i + \mathbf{m}_{i+1}$$
$$\mathbf{m}_{i+1} = \alpha_{i+1} \mathbf{m}_i - \eta_t \mathbf{P}_i \nabla \mathcal{L}(W_i; x_i). \tag{20}$$

This formulation is equivalent to using preconditioning the momentum GD. In fact, preconditioning means that the momentum term is an associative memory that learns how to compress the mappings between $\mathbf{P}_i$ and the gradient term $\nabla \mathcal{L}(W_i; x_i)$. While any reasonable choice (e.g., random features) of preconditioning can improve the expressivity of the initial version of GD with momentum per se is a value-less memory (i.e., mapping all gradients to a single value), the above perspective gives more intuition about what preconditioning are more useful. That is, the momentum acts as a memory that aims to map gradients to their corresponding values, and so a function of gradients (e.g., information about Hessian) can provide the memory with a more meaningful mappings.

**Extension: More Expressive Objectives.** As discussed by Behrouz et al. [58], optimizing an inner objective of dot-product similarity results in Hebbian-like update rule, which can cause the memory to be less effective. A natural extension of this internal objective is to use $\ell_2(\cdot)$ regression loss (for measuring the corresponding key-value mapping fitness) and minimize the loss function $\|\mathbf{m} \nabla \mathcal{L}(W_i; x_i)^\top - \mathbf{P}_i\|_2^2$, resulting in the update rule of:

$$W_{i+1} = W_i + \mathbf{m}_{i+1}, \tag{21}$$
$$\mathbf{m}_{i+1} = \left( \alpha_{i+1} \mathbf{I} - \nabla \mathcal{L}(W_i; x_i)^\top \nabla \mathcal{L}(W_i; x_i) \right) \mathbf{m}_i - \eta_t \mathbf{P}_i \nabla \mathcal{L}(W_i; x_i), \tag{22}$$

This update is based on delta-rule [64] and so it allows the memory (momentum) to better manage its limited capacity and better memorize the series of past gradients.

**Extension: More Expressive Memory.** As discussed earlier, momentum can be viewed as a meta memory model that uses a linear layer (i.e., matrix-valued) to compress the past gradient values. Due to the linear nature of momentum, only linear functions of past gradients can be learned by its internal objective. To increase the learning capacity of this module, one alternative is to use alternative powerful persistent learning modules: i.e., replacing a linear matrix-valued memory for momentum with an MLP. Therefore, momentum as the a memory for the past gradients, has more capacity to capture the underlying dynamics of the gradients. To this end, we extend the formulation in Equation 17 as:

$$W_{i+1} = W_i + \mathbf{m}_{i+1}(\mathbf{u}_i), \quad \text{and} \quad \mathbf{m}_{i+1} = \alpha_{i+1} \mathbf{m}_i - \eta_t \nabla \mathcal{L}^{(2)}(\mathbf{m}_i; \mathbf{u}_i, \mathbf{I}), \tag{23}$$

where $\mathbf{u}_i = \nabla \mathcal{L}(W_i; x_i)$ and $\nabla \mathcal{L}^{(2)}(\cdot)$ is the internal objective of momentum (e.g., dot product similarity $\langle \mathbf{m}(\mathbf{u}_i^\top), \mathbf{1} \rangle$). We refer to this variant as Deep Momentum Gradient Descent (DMGD).

**Extension: None Linear Outputs.** Building upon the above perspective, in which we see the momentum as a neural architecture, one common technique to enhance the representation power of momentum memory module is to use non-linearity on top of its output [28, 65]. That is, we re-formulate Equation 23 as:

$$W_{i+1} = W_i + \sigma(\mathbf{m}_{i+1}(\mathbf{u}_i)), \quad \text{and} \quad \mathbf{m}_{i+1} = \alpha_{i+1} \mathbf{m}_i - \eta_t \nabla \mathcal{L}^{(2)}(\mathbf{m}_i; \mathbf{u}_i, \mathbf{I}), \tag{24}$$

where $\sigma(\cdot)$ is an arbitrary non-linearity. As an example, we let $\sigma(\cdot) = \texttt{Newton-Schulz}(\cdot)$, where $\texttt{Newton-Schulz}(\cdot)$ is the iterative Newton-Schulz method [66], and $\mathbf{m}(\cdot)$ be a linear layer; the resulted optimizer is equivalent to Muon optimizer [34].

**Going Beyond Simple Backpropagation.** As discussed earlier in Section 2.1, the pre-training process and backpropagation is a form of associative memory, where input data is mapped to the surprised caused by its predicted output $\nabla_{y_t}\mathcal{L}(W_t; x_t)$:

$$W_{t+1} = W_t - \eta_{t+1}\nabla_{W_t}\mathcal{L}(W_t; x_t) = W_t - \eta_{t+1}\nabla_{y_t}\mathcal{L}(W_t; x_t) \otimes x_t, \quad \text{where } x_t \sim \mathcal{D}_{\text{train}}, \quad (25)$$

which from the associative memory perspective is equivalent to one step of gradient descent in optimization process of:

$$\min_W \ \langle Wx_t, \nabla_{y_t}\mathcal{L}(W_t; x_t)\rangle. \tag{26}$$

As we discussed in Appendix D, the above formulation cause ignoring the dependencies of data samples like $x_t$. To extend it to a more powerful formulation where it also consider the dependencies of data points (which is extremely important when we use optimizer in the token space as they are not independent), we use $L_2$ regression objective with one step of gradient descent as follows:

$$\min_W \ \|Wx_t - \nabla_{y_t}\mathcal{L}(W_t; x_t)\|_2^2. \tag{27}$$

This formulation results in a new variant of gradient descent, which can be simplified as follows:

$$W_{t+1} = W_t\left(\mathbf{I} - x_tx_t^\top\right) - \eta_{t+1}\nabla_{W_t}\mathcal{L}(W_t; x_t) \tag{28}$$

$$= W_t\left(\mathbf{I} - x_tx_t^\top\right) - \eta_{t+1}\nabla_{y_t}\mathcal{L}(W_t; x_t) \otimes x_t, \qquad \text{where } x_t \sim \mathcal{D}_{\text{train}}, \tag{29}$$

Later, we use this optimizer as the internal optimizer of our HOPE architecture.

## 3 HOPE: A Self-Referential Learning Module with Continuum Memory

Existing architectural backbones consist of (1) a *working memory* module (e.g., attention), which is responsible to actively fuse the information across sequence length, and (2) a feed-forward layer (e.g., MLP) that fuse information across features and acts as the persistent memory or knowledge storage of pre-training phase. From the NL perspective, pre-training is the phase that the most outer level of the learning module is updated over its *limited* context flow. Accordingly, in the continual setup, such pre-training phase is also rarely updated over time, and so its corresponding knowledge storage needs to rarely be updated over time. Given this intuition, we extend the traditional view-point of long-term/short-term memory system and suggest a knowledge storage feed-forward for each level (frequency domain).

Given the definition of frequency and arbitrary chosen objective $\mathcal{L}$, Continuum Memory System (CMS) is formalized as a chain of MLP blocks $\texttt{MLP}^{(f_1)}(\cdot), \ldots, \texttt{MLP}^{(f_k)}(\cdot)$, each of which associated with a chunk size of $C^{(\ell)} := \frac{\max_t C^{(t)}}{f_\ell}$ such that given input $x = \{x_1, \ldots, x_T\}$ the output of the chain is calculated as (we disregard normalizations for the sake of clarity):

$$y_t = \texttt{MLP}^{(f_k)}(\texttt{MLP}^{(f_{k-1})}(\cdots \texttt{MLP}^{(f_1)}(x_t))), \tag{30}$$

where the parameters of $\ell$-th MLP block, i.e., $\boldsymbol{\theta}^{(f_\ell)}$, are updated every $C^{(\ell)}$ steps:

$$\boldsymbol{\theta}_{i+1}^{(f_\ell)} = \boldsymbol{\theta}_i^{(f_\ell)} - \begin{cases} \sum_{t=i-C^{(\ell)}}^{i} \eta_t^{(\ell)} f(\boldsymbol{\theta}_t^{(f_\ell)}; x_t) & \text{if } i \equiv 0 \pmod{C^{(\ell)}}, \\ 0 & \text{otherwise.} \end{cases} \tag{31}$$

In Appendix C, we discuss different variants of this formulation, including fully nested MLP layers. Here $\eta_t^{(\ell)}$ are learning rates corresponds to $\boldsymbol{\theta}^{(f_\ell)}$, and $f(\cdot)$ is the error component of an arbitrary optimizer (e.g., $\nabla\mathcal{L}(\boldsymbol{\theta}_t^{(f_\ell)}; x_t)$ in gradient descent). The MLP block in conventional Transformers architecture [27] is a special instance of this formulation, where $k = 1$ and frequency is zero (after pre-training). It is notable that Equation 31 provides an important interpretation: parameters $\boldsymbol{\theta}_t^{(f_\ell)}$ are responsible for compressing their own context into the their parameters and so they are a representative of abstract knowledge of their context.

**HOPE.** We further present a self-referential learning module based on Titans [28] and our variant of gradient descent in Appendix B. Combining this self-referential sequence model with continuum memory system results in HOPE architecture.

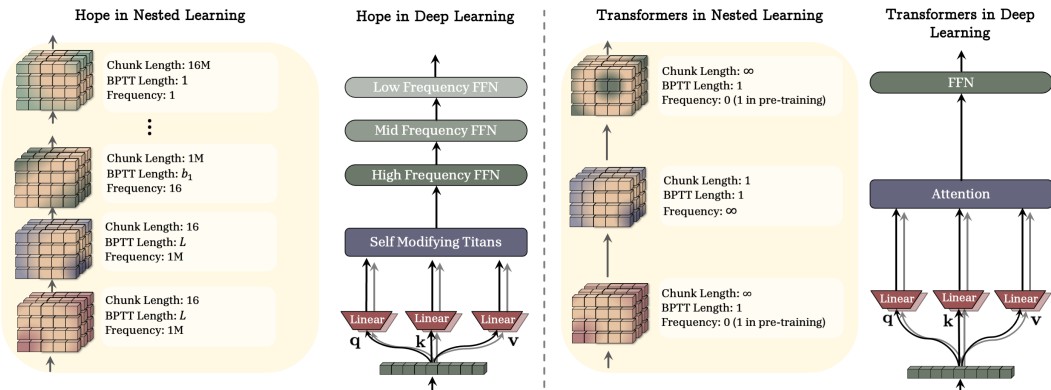

Figure 3: A comparison of Hope architectural backbone with Transformers (Normalization and potential data-dependent components are removed for the sake of clarity). It is notable that while self-modifying Titans is a replacement for the attention block, CMS is a replacement for the static MLP block.

Table 1: Performance of HOPE and baselines on language modeling and common-sense reasoning tasks. Hybrid models are marked with *.

| Model | Wiki. ppl ↓ | LMB. ppl ↓ | LMB. acc ↑ | PIQA acc ↑ | Hella. acc_n ↑ | Wino. acc ↑ | ARC-e acc ↑ | ARC-c acc_n ↑ | SIQA acc ↑ | BoolQ acc ↑ | Avg. ↑ |
|---|---|---|---|---|---|---|---|---|---|---|---|
| | | | | | 760M params / 30B tokens | | | | | | |
| Transformer++ | 25.21 | 27.64 | 35.78 | 66.92 | 42.19 | 51.95 | 60.38 | 32.46 | 39.51 | 60.37 | 48.69 |
| RetNet | 26.08 | 24.45 | 34.51 | 67.19 | 41.63 | 52.09 | 63.17 | 32.78 | 38.36 | 57.92 | 48.46 |
| DeltaNet | 24.37 | 24.60 | 37.06 | 66.93 | 41.98 | 50.65 | 64.87 | 31.39 | 39.88 | 59.02 | 48.97 |
| TTT | 24.17 | 23.51 | 34.74 | 67.25 | 43.92 | 50.99 | 64.53 | 33.81 | 40.16 | 59.58 | 47.32 |
| Samba* | 20.63 | 22.71 | 39.72 | 69.19 | 47.35 | 52.01 | 66.92 | 33.20 | 38.98 | 61.24 | 51.08 |
| Titans (LMM) | 20.04 | 21.96 | 37.40 | 69.28 | 48.46 | 52.27 | 66.31 | 35.84 | 40.13 | 62.76 | 51.56 |
| HOPE (ours) | 20.53 | 20.47 | 39.02 | 70.13 | 49.21 | 52.70 | 66.89 | 36.05 | 40.71 | 63.29 | 52.26 |
| | | | | | 1.3B params / 100B tokens | | | | | | |
| Transformer++ | 18.53 | 18.32 | 42.60 | 70.02 | 50.23 | 53.51 | 68.83 | 35.10 | 40.66 | 57.09 | 52.25 |
| RetNet | 19.08 | 17.27 | 40.52 | 70.07 | 49.16 | 54.14 | 67.34 | 33.78 | 40.78 | 60.39 | 52.02 |
| DeltaNet | 17.71 | 16.88 | 42.46 | 70.72 | 50.93 | 53.35 | 68.47 | 35.66 | 40.22 | 55.29 | 52.14 |
| Samba* | 16.13 | 13.29 | 44.94 | 70.94 | 53.42 | 55.56 | 68.81 | 36.17 | 39.96 | 62.11 | 54.00 |
| Titans (LMM) | 15.60 | 11.41 | 49.14 | 73.09 | 56.31 | 59.81 | 72.43 | 40.82 | 42.05 | 60.97 | 56.82 |
| HOPE (ours) | 15.11 | 11.63 | 50.01 | 73.29 | 56.84 | 60.19 | 72.30 | 41.24 | 42.52 | 61.46 | 57.23 |

## 4   Experiments

For the sake of space, in the main paper, we report the results of the HOPE's evaluation on language modeling, and common-sense reasoning, tasks. However, we report an extensive set of results, including on experiments on optimizers, emergence of in-context learning, continual learning abilities of HOPE, ablation studies, long-context tasks, etc. in the appendix. Details about the experimental setups and other used datasets are in Appendix F

**Language Modeling and Common-sense Reasoning.** We follow recent sequence modeling studies [28, 67, 68] and report the results of HOPE and baselines with size of 760M, and 1.3B on language modeling and also commonsense reasoning downstream tasks [69–75]. These results are reported in Table 1. HOPE demonstrate a very good perfomance across all the scales and benchmark tasks, outperforming both Transformers and recent modern recurrent neural networks, including DeltaNet [63] and Titans [28]. Comparing HOPE to Titans and DeltaNet, we can see that dynamically changing the key, value, and query projections based on the context as well a deep memory module can result in a model with lower perplexity and higher accuracy in benchmark results.

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

# A  Additional Related Work

In this section, we provide a more comprehensive review of literature that are connected to our work:

**Modern Deep Learning Architectures.** Recent research has focused on developing efficient recurrent alternatives to Transformers to mitigate their quadratic computational cost and limitations in modeling long contexts, primarily driven by the faster inference and training capabilities of recurrent architectures [76]. Initial models, including RetNet [61], RWKV [77], and S5 [78], employed data-independent transition matrices with Hebbian-like update rules. Subsequent approaches began integrating input-dependent parameters into these linear architectures (e.g. SSMs [79, 80], RWKV6 [81]) or adopted more expressive memory updating mechanisms based on the delta rule [59, 63, 82–84]. More recent advancements have extended these memory architectures to deeper models, utilizing delta-rule-like [65] or momentum-based [28] update rules. To further improve delta-rule-based sequence models, Siems et al. [85] proposed employing multiple gradient descent updates per token, leading to enhanced expressiveness in state tracking tasks.

**Fast Weight Programs.** The conceptualization of recurrence as key-value associative memory systems originates from Hopfield networks [86] and fast weight programmers, where dynamic fast programs function as writable memory within recurrent neural networks [62, 63, 87]. Among the learning paradigms for these systems, the Hebbian [88] and delta [64] rules are prominent and have been extensively investigated in the literature [62, 63, 83, 84, 89–91]. Also, recently, the duality of meta-learning perspective and the architecture has been in the core of attention of several studies [58, 63, 92, 93]. Furthermore, our HOPE, can be seen as the generalization of SRWM [94], where instead of a matrix valued memory, HOPE uses a neural networks and also its update rule is generalized using: (1) a data-dependent gating mechanism that helps to forget the past data; (2) We update the memory using gradient descent with momentum, where SRWM is equivalent to update the memory using a simple gradient descent. Also, while SRWM is not parallelizable, we present a parallelizable training algorithm for our HOPE.

# B  Reformulating Modern Architectures as Learning modules

Modern sequence models such as Transformers [27] and recurrent models [28, 60, 63, 65] are the backbones of recent advances in language models. Recently, the equivalency of such models with associative memories that aim to learn a mapping from keys to values from data have been studied in different settings and objectives [58, 59, 65, 92]. Particularly, we focus on the general framework of Miras [58], which defines associative memory as Definition 1 and optimizes the internal objective (called "attentional bias") with a choice of optimization algorithm on an arbitrary class of functions (i.e., memory architecture). While this formulation alone indicates that the well-known architectures are instances of nested systems of associative memory, next, we review this equivalency for some learning rules and architectures.

From now on, we assume that keys $\{k_i\}_{i=1}^L$, values $\{v_i\}_{i=1}^L$, and queries $\{q_i\}_{i=1}^L$ are given: they often are defined as the projections of the input, i.e.,

$$k_t = x_t W_k, \qquad v_t = x_t W_v, \qquad q_t = x_t W_q. \tag{32}$$

In this design, since projection parameters (i.e., $W_k$, $W_v$, and $W_q$) are optimized in a lower frequency level, the sequence model component (e.g., self-attention) has a higher frequency and so the learning process of the associative memory happens in a lower level. Accordingly, for the sake of clarity, we only discuss the higher frequency level (i.e., the internal learning process of the associative memory).

**Softmax Attention.** From the associative memory viewpoint: given keys $\{k_i\}_{i=1}^L$, values $\{v_i\}_{i=1}^L$, and queries $\{q_i\}_{i=1}^L$, `Softmax` attention block [27, 95] can be reformulated as a non-parametric solution to the $\ell_2(\cdot)$ regression objective with Nadaraya-Watson estimators [96, 97]:

$$\mathcal{M}^* = \arg\min_{\mathcal{M}} \sum_{i=1}^L \mathsf{s}(k_i, q)\|v_i - \mathcal{M}\|_2^2 = \sum_{i=1}^L \frac{\mathsf{s}(k_i, q)}{\sum_{j=1}^L \mathsf{s}(k_j, q)} v_i, \tag{33}$$

where $L$ is the sequence length [65]. This formulation optimizes the memory $\mathcal{M}(\cdot)$ with respect to the entire context; however, one design choice can be to limit the optimization process to the past $c$

tokens, resulting in:

$$\mathcal{M}^* = \arg\min_{\mathcal{M}} \sum_{i=t-c+1}^{t} \mathbf{s}(\boldsymbol{k}_i, \boldsymbol{q}_i) \|\boldsymbol{v}_i - \mathcal{M}\|_2^2 = \sum_{i=t-c+1}^{t} \frac{\mathbf{s}(\boldsymbol{k}_i, \boldsymbol{q})}{\sum_{j=t-c+1}^{t} \mathbf{s}(\boldsymbol{k}_j, \boldsymbol{q})} \boldsymbol{v}_i, \qquad (34)$$

which is equivalent to the sliding window attention (SWA). Therefore, attention and its more expressive variants [92] also are instances of Definition 1, when instead of gradient descent or other parametric methods, we find the optimal non-parametric solution to the mapping.

**RNNs with Hebbian Rule.** The first generation of modern recurrent architectures (e.g., Linear attention [60], RetNet [61], RWKV [77], lightening attention [98]) are based on Hebbian-like learning rules [88]. For this class of models, the inner objective to measure the quality of mapping between keys and values is the dot-product similarity. That is, given a matrix-valued memory $\mathcal{M} \in \mathbb{R}^{d \times n}$, keys and values $\boldsymbol{k}, \boldsymbol{v} \in \mathbb{R}^d$, objective $\tilde{\mathcal{L}}(\mathcal{M}; \boldsymbol{k}_t, \boldsymbol{v}_t) := -2\langle \mathcal{M}\boldsymbol{k}_t, \boldsymbol{v}_t \rangle$, and a kernel $\phi(\cdot)$, we optimize the equivalent associative memory optimization problem (see Definition 1) with gradient descent and weight decay, resulting in:

$$\mathcal{M}_t = \alpha_t \mathcal{M}_{t-1} - \eta \underbrace{\nabla_{\mathcal{M}_{t-1}} \tilde{\mathcal{L}}(\mathcal{M}_{t-1}; \phi(\boldsymbol{k}_t), \boldsymbol{v}_t)}_{-\boldsymbol{v}_t \phi(\boldsymbol{k}_t^\top)} = \alpha_t \mathcal{M}_{t-1} + \eta_t \, \boldsymbol{v}_t \phi\left(\boldsymbol{k}_t^\top\right), \qquad (35)$$

which recovers the original linear attention recurrence [60]. Given different settings for $\alpha_t$ (i.e., either is 1, learnable, channel-wise, and/or input-dependent) and also $\phi(\cdot)$ (i.e., identity, polynomial kernels, etc.), the above recurrence recovers different variants of linear attention with Hebbian rule [60, 61, 81, 99–101]. Therefore, the variants of linear attention with Hebbian rule can be reformulated as the process of an optimization problem, in which the memory aims to learn the mapping between keys and values based on dot-product similarity objective, with gradient descent.

**RNNs with Delta Rule.** To improve the memory management and to enhance the memory capacity of the above group, several studies suggest replacing Hebbian rule with Delta rule as the learning algorithm in recurrent neural networks [63], resulting in models such as DeltaNet [63], Longhorn [59], and RWKV7 [82]. When letting $\mathcal{M} \in \mathbb{R}^{d \times n}$, delta rule is equivalent to optimizing MSE objective $\tilde{\mathcal{L}}_t = \|\mathcal{M}_t \boldsymbol{k}_t - \boldsymbol{v}_t\|_2^2$ with $\text{Ret}_t(\mathcal{M}, \mathcal{M}_{t-1}) = \|\mathcal{M}_t - \mathcal{M}_{t-1}\|_F^2$ as local retention, and stochastic gradient descent as the optimizer:

$$\mathcal{M}_t = \mathcal{M}_{t-1} - \eta_t \underbrace{\nabla_{\mathcal{M}_{t-1}} \tilde{\mathcal{L}}(\mathcal{M}_{t-1}; \phi(\boldsymbol{k}_t), \boldsymbol{v}_t)}_{(\mathcal{M}_{t-1}\boldsymbol{k}_t - \boldsymbol{v}_t)\boldsymbol{k}_t^\top} = \left(\mathbf{I} - \eta_t \boldsymbol{k}_t \boldsymbol{k}_t^\top\right) \mathcal{M}_{t-1} + \eta_t \, \boldsymbol{v}_t \boldsymbol{k}_t^\top. \qquad (36)$$

Using other forms of retention gates (e.g., $\text{Ret}_t(\mathcal{M}, \mathcal{M}_{t-1}) = \|\mathcal{M}_t - \alpha_t \, \mathcal{M}_{t-1}\|_F^2$), optimization algorithms with weight decay (e.g., regularizing with $\|\mathcal{M}_t\|_q^q$ for a given $q > 0$), multiple steps of gradient descent, and/or different formulations of learnable parameters such as $\eta_t$ and $\alpha_t$ can result in diverse variants of delta rule [58, 59, 65, 67, 82, 85, 91, 92]. Therefore, Delta rule and its variants are all instances of an optimization problem, in which the model aims to learn a mapping between keys and values based on the $L_2$-regression objective.

**Beyond Conventional Learning Rules: Omega, Oja's, and Non-Euclidean Learning Rules.** More recently, there have been growing interests in designing architectures from the associative memory perspective (see Definition 1) and use more complex internal objectives, and/or optimization algorithms, resulting in learning algorithms beyond Delta and Hebbian rules [58, 68, 102–104]. More specifically, to enhance the stability of Hebbian rule (discussed in Equation 35), Irie et al. [102] introduced OjaNet based on Oja's rule [105] with the following recurrence:

$$\mathcal{M}_t = \alpha_t \mathcal{M}_{t-1} + \eta_t \, \boldsymbol{v}_t \left(\phi(\boldsymbol{k}_t)^\top - \mathcal{M}_{t-1}^\top \boldsymbol{v}_t\right). \qquad (37)$$

In the associative memory formulation (as in Definition 1), this recurrence can simply be reformulated as one step of gradient descent as:

$$\mathcal{M}_t = \mathcal{M}_{t-1} - \eta_t \underbrace{\nabla_{\mathcal{M}_{t-1}} \tilde{\mathcal{L}}(\mathcal{M}_{t-1}; \phi(\boldsymbol{k}_t), \boldsymbol{v}_t)}_{\mathcal{M}_{t-1}^\top \boldsymbol{v}_t - \boldsymbol{v}_t \phi(\boldsymbol{k}_t)^\top}, \qquad (38)$$

where $\tilde{\mathcal{L}}(\mathcal{M}; \boldsymbol{k}_t, \boldsymbol{v}_t) = -2\langle \mathcal{M}\boldsymbol{k}_t, \boldsymbol{v}_t\rangle + \|\mathcal{M}^\top \boldsymbol{v}_t\|_2^2$ and $\phi(\cdot)$ is a kernel [102, 106]. Although this design enhances the Hebbian learning rule by enforcing a unit-norm constraint for the single-neuron, it has been reported to empirically underperform models based on Delta learning rule [102]. To further enhance the Delta rule through the design of more expressive objectives, recently, Behrouz et al. [58] suggested going beyond Euclidean spaces and use $L_p = \|\cdot\|_p^p$ norm for the internal regression objective, showing better empirical performance and robustness in long context tasks compared to Delta rule and its variants.

While the majority of learning rules are online update mechanisms–meaning that at each state, the models only need to keep the memory and the *current* (batch of) input–Omega rule [68] suggest an update rule based on a set of past (batches of) inputs (or all inputs). More specifically, given a memory $\mathcal{M}$ with an arbitrary structure, keys and values $\boldsymbol{k}, \boldsymbol{v} \in \mathbb{R}^d$, an arbitrary objective $\tilde{\mathcal{L}}(\mathcal{M}; \boldsymbol{k}_t, \boldsymbol{v}_t)$, and a kernel $\phi(\cdot)$, Omega rule is defined as:

$$\mathcal{M}_t = \alpha_t \, \mathcal{M}_{t-1} - \sum_{i=t-c+1}^{t} \gamma_{t,i} \, \tilde{\mathcal{L}}(\mathcal{M}_t; \phi(\boldsymbol{k}_i), \boldsymbol{v}_i), \tag{39}$$

where $c \geq 1$ is the local window of cached inputs. Note that in the special case of $\gamma_{t,i} = 1$ and $c$ equal to the entire context length, the optimal solution of the above design collapses into an online case, where the update rule only depends on the current state and the current input [103]. For further discussion with more details about representing architectures as associative memories and so an optimization problem, we refer the reader to Behrouz et al. [58].

**A Note on Gating in Modern Sequence Models.** One of the recent architectural changes in modern language models is the gating of a linear layer's output with the output of the sequence model. Despite significant improvement resulted by this method, it is still unclear that how it enhances the performance. The main difference between feedforward network and modern recurrent memory modules (e.g., linear attention [60] or deep memory modules [28]) when their initial state of the memory is meta-learned, is the second level in memory modules that perform in-context learning and adapt its state with the context. From this viewpoint, when the initial value of the memory is not meta-learned, it only relies on the in-context adaption of the memory and so there is no persistent memory system that stores the knowledge of pre-training in this block. Therefore, when the initial value of memory is not meta-learned, which is common in earlier variants of linear transformers, the gating of linear attention acts as a persistent memory and the initialization of the memory module.

## C  Details on initialization of CMS

As discussed earlier, different levels might have different process of knowledge transfer. Accordingly, while the above formulation suggests a spectrum of memory systems in different levels and so with different frequencies, their connections can vary based on the design. In the following, we discuss some potential variants:

**Nested Continuum Memory Systems.** The first variant is a fully nested continuum memory system, in which the initial state of the MLP block in level $s + 1$ is meta-learned in level $s$. This design allows for higher-order in-context learning ability, where each of the levels has its own context flow and re-initialized after the end of the context. More specifically, given an arbitrary $1 \leq s \leq k$,

$$\boldsymbol{\theta}_0^{(f_{s+1})} = \arg\min_{\Phi} \; \mathbb{E}_{\mathcal{T} \sim \mathcal{C}^{(s)}} \left[ \ell(\Theta, \mathcal{T}; \Phi) \right], \tag{40}$$

where $\mathcal{C}^{(s)}$ is the context length of the MLP block in $s$-th level. Following this design, at the end of the optimization process of each block (i.e., after $\lceil C^{(s)}/C^{(s+1)} \rceil$ steps.) the value of the memory will be re-initialized to $\boldsymbol{\theta}_0^{(f_{s+1})}$. Note that the update mechanism of each block in its own level remain unchanged (i.e., Equation 31).

**Sequential Continuum Memory Systems.** In the second variant, the MLP blocks are located sequentially (i.e., the output of the MLP block in level $s$ is the input for the MLP block in level $s + 1$) and also the initial state of MLP blocks are all connected through backpropagation in the lowest

frequency level. Given an arbitrary $1 \leq s \leq k$,

$$\boldsymbol{\theta}_0^{(f_s)} = \arg\min_{\Phi} \ \mathbb{E}_{\mathcal{T} \sim \mathcal{C}^{(1)}} \left[ \ell(\Theta, \mathcal{T}; \Phi) \right], \tag{41}$$

where $\mathcal{C}^{(1)}$ is the context length of the MLP block in the lowest frequency level. Since the initial state of all memories are meta-learned in the lowest frequency, the most persistent knowledge of all components is the compression of the same context flow.

**Independent (Head-wise) Continuum Memory Systems.** In this variant, we keep the knowledge transfer process in Equation 41, but change the output computation in Equation 30. While the previous formulation designs the memory system as a sequence of blocks, and so making their input/out dependent to each other, this variant uses independent blocks with different context length and then combine them using an aggregation process:

$$\boldsymbol{y}_t = \texttt{Agg}\left( \texttt{MLP}^{(f_k)}(\boldsymbol{x}_t), \texttt{MLP}^{(f_{k-1})}(\boldsymbol{x}_t), \cdots, \texttt{MLP}^{(f_1)}(\boldsymbol{x}_t) \right). \tag{42}$$

The above $\texttt{Agg}(\cdot)$ is an arbitrary function that aggregates all the inputs to compute the output. For example, one straightforward and simple design choice is to use a learnable weighted sum of the input.

**CMS Design Helps with Continual Learning.** Based on the design of CMS, a fair question is to ask: Why and how CMS can help with longer context length and generally continual learning. Here, we provide a simple answer to this question: Viewing MLP blocks in CMS as the storage of model's knowledge catastrophic forgetting can happen when we update a block and as its result, the old knowledge stored in its parameters are forgotten. In CMS design, however, when updating an arbitrary block of $\texttt{MLP}^{(f_s)}(\cdot)$ for some $1 \leq s \leq k$, the potentially forgotten knowledge from $\texttt{MLP}^{(f_s)}(\cdot)$ is still stored in other components such as $\texttt{MLP}^{(f_{s'})}(\cdot)$, where $s' < s$. Also, in this case (i.e., the knowledge is already forgotten from $\texttt{MLP}^{(f_s)}(\cdot)$ but it is still in $\texttt{MLP}^{(f_{s'})}(\cdot)$ for $s' < s$) the knowledge transfer through backpropagation (for their initial state) can circle back the knowledge to $\texttt{MLP}^{(f_s)}(\cdot)$, resulting in a loop through time dimension, and so hardly forgetting important knowledge.

**Is CMS Efficient Enough?** A common concern when updating the parameters of a model in a continual manner is its efficiency. Therefore, a fair question is to ask if CMS causes significant computational overhead for the model. To answer this question, let us recall from Appendix B that modern recurrent neural networks are also continually updating a subset of their parameters (i.e., their memory state). These parameter updates, however, take advantage of sequence parallelization as well as updating only a small number of parameters. To this end, for CMS, we highlight two points:

- In the CMS design, at each time, updates are restricted to blocks approaching their scheduled update time (based on their frequency). As a simple example, consider a Transformers but with replacing its MLP blocks with CMS. Let the model have $L_{\text{layer}}$ layers, 4 levels of MLP blocks in CMS with highest frequency of $\hat{f}$, and hidden dimension of $d_{\text{in}}$. On average, the update cost is for $\mathcal{O}\left( \frac{1}{\hat{f}} \times \frac{L_{\text{layer}}}{5} \times d_{\text{in}}^2 \right)$ of parameters, which consists of only a small number of parameters at each time.

- The update mechanism of Equation 31, not only helps with the enhancing the persistent memory of the model, but it also unlocks the sequence parallelization for higher frequency levels. More specifically, for input $\boldsymbol{x}_i$ when $i \not\equiv 0 \pmod{C^{(\ell)}}$ there is no sequential process inside the chunk and so all the computations for tokens correspond to different values of $i \not\equiv 0 \pmod{C^{(\ell)}}$ can be done in parallel. The details of such training algorithm is the same as the training procedure in Behrouz et al. [28], Sun et al. [65].

Therefore, in summary, CMS can be fast in practice, mainly due to the fact that it updates only small number of parameters at each time, and also its design unlocks sequence parallelization.

# D    Reformulating Modern Optimizers as Instances of NL

In this section, we start with viewing backpropagation process and optimizing a neural network from the associative memory and data compression perspective. Then, we discuss how variants such as

momentum-based optimizers are instances of nested associative memory systems. Finally, we discuss alternative methods leading to deep optimizers with higher expressive power from the associative memory perspective.

## D.1 Backpropagation as an Associative Memory

Updating the weights of a neural network through backpropagation [29, 107] has been the critical component of training large-scale deep neural networks. Intuitively, in this optimization process, first, the error of the model's output with respect to target is calculated, and then each layer is updated based on its contribution to this error. This section aims to explain this process through the lens of associative memory and discuss how it fits within the nested learning paradigm. For the sake of clarity and simplicity, we assume a deep MLP model, but all the derived formulations in the following can simply be adapted to other architectures as well. Given an MLP with $L$ layers parameterized with $\{W_\ell \cdot + b_\ell\}_{\ell=1}^L$, the required gradients in backpropagation are computed as:

$$\frac{\partial \mathcal{L}}{\partial W_\ell} = \delta_\ell \, \hat{x}_{\ell-1}^\top, \qquad \text{and} \qquad \delta_\ell = \underbrace{J_{\phi_\ell}(z_\ell)^\top \left(W_{\ell+1}^\top \delta_{\ell+1}\right)}_{\text{local output surprise for layer } \ell}, \tag{43}$$

where $z_\ell = W_\ell \, \hat{x}_{\ell-1} + b_\ell$ is pre-activation, and so $\hat{x}_\ell = \phi_\ell(z_\ell)$ is the output of $\ell$-th layer, $\phi_\ell(\cdot)$ is its non-linearity, and $J_{\phi_\ell}(\cdot)$ is the Jacobian. Therefore, the update of the $\ell$-th layer with gradient descent is computed as:

$$W_{\ell_{t+1}} = W_{\ell_t} - \eta_{\ell_{t+1}} \, \delta_\ell \, \hat{x}_{\ell-1}^\top . \tag{44}$$

Here, $\hat{x}_{\ell-1}$ is the input of the layer and $\delta_\ell$ measures the local error signal for layer $\ell$ or equivalently is a metric that measures the surprise of layer $\ell$'s output given its input. Similar to our example in Section 2.1, we can write Equation 44 as:

$$W_{\ell_{t+1}} = \arg\min_W \langle W\hat{x}_{\ell-1}, \delta_\ell \rangle + \frac{1}{2\eta_{\ell_{t+1}}} \|W - W_{\ell_t}\|_F^2, \tag{45}$$

which is an associative memory module that aim to map the input of each layer $\hat{x}_{\ell-1}$ to its local error signal, $\delta_\ell$ (see Definition 1). That is, the above formulation implies that *the training process of a neural network with gradient descent and backpropagation can be viewed as a compression process*, in which each layer stores the mappings between its input and the corresponding local error signal.

## D.2 Momentum-based Optimizers as Associative Memories

Momentum-based optimizers are the major components of modern machine learning models' training [33, 34, 108]. To explain momentum-based optimizers as associative memories, let us start from a simple gradient descent algorithm:

$$W_{t+1} = W_t - \eta_t \nabla_{W_t} \mathcal{L}(W_t; x_{t+1}), \tag{46}$$

which updates the current state of the weights based on the momentary gradient (surprise). This update rule does not incorporate the previous tokens and also the loss landscape that have been traversed so far, resulting in slower (or less robust) convergence in many scenarios. To fix this, momentum-based gradient descent methods incorporate an Exponential Moving Averages (EMAs) of past gradients:

$$W_{\ell_{t+1}} = W_{\ell_t} + m_{\ell_{t+1}}$$
$$m_{\ell_{t+1}} = \alpha_{\ell,t+1} m_{\ell_t} - \eta_{\ell,t+1} \nabla_{W_{\ell_t}} \mathcal{L}\left(W_{\ell_t}; x_{t+1}\right) = \alpha_{\ell,t+1} m_{\ell_t} - \eta_{\ell,t+1} \, \delta_\ell \, \hat{x}_{\ell-1}^\top, \tag{47}$$

where matrix (or vector) $m_t$ is the momentum at state $t$ and $\alpha_t$ and $\eta_t$ are (adaptive) learning and momentum rates, respectively, and $\delta_\ell$ and $\hat{x}_{\ell-1}$ are defined the same as in Equation 43. Similar to Equation 45 and one of the examples in Section 2.1, assuming $\alpha_{t+1} = 1$, the momentum term can be viewed as the result of optimizing the following objective with gradient descent:

$$\min_m \langle m \, \hat{x}_{\ell-1}, \delta_\ell \rangle. \tag{48}$$

The case of $\alpha_{t+1} \neq 1$ is equivalent to GD on the above minimization plus an $\ell_2$-regularization on the momentum term. Thus, momentum can indeed be viewed as an associative memory module

that learns how to compress the past gradients of the objective into its parameters. Contrary to Equation 45, which was a simple 1-level associative memory and the update was directly applied to the memory, here the state of the momentum determines the update for the weights. In other words, it is a 2-level optimization procedure, in which the inner-loop learns the momentum and the outer-loop uses the state of the momentum to update the weights.

From this perspective, we can generalize the definition of momentum from EMAs to any arbitrary associative memory module that aims to compress the past gradients or maps the input of each token to its corresponding local error. This generalized momentum can be expressed as:

$$W_{\ell_{t+1}} = W_{\ell_t} + m_{\ell_{t+1}},$$ (49)

(50)

where $m_\ell$ is the solution of the following associative memory, optimized by gradient descent:

$$\min_m \tilde{\mathcal{L}}\left(m; \, \hat{x}_{\ell-1}, -\delta_\ell\right).$$ (51)

Here, the objective $\tilde{\mathcal{L}}(\cdot)$ is different from the original objective of the problem at hand, and $\tilde{\mathcal{L}}(\cdot)$ is the objective that defines the momentum and measures the quality of its mappings. In fact, the momentum term in this formulation aims to adapt in-context (recall that the context of the momentum is the gradients) to the local error rates based on the input of the layer. Most popular optimizers are formulated as element-wise update rule (for computational efficiency reasons) and so we first explore the element-wise associative memory formulation of momentum and connect it to popular optimizers such as Adam [33]. Showing that Adam can be viewed as the optimal associative memory to the $L_2$-regression objective that aims to predict the variance of gradients, we discuss other similar algorithms such as RMSProp [109], SignSGD and its momentum-based variants [110], NAdam [111], AMSGrad [112], RAdam [113], and Lion [114] are also instances of an associative memory that aims to compress the gradients. We then go beyond element-wise formulation and show that AdaGrad [108] is also an associative memory module. Due to the connection of AdaGrad with optimizers such as Shampoo [35] and Soap [36]–i.e., as the approximation of the preconditioning term–we then conclude that all these optimizers can be re-formulated as associative memory. Next, we discuss another class of optimizers based on preconditioning and reformulate them from NL's perspective in more details:

**Preconditioning and Approximation of Hessian.** Another class of algorithms is preconditioning algorithms where the idea is to approximate Hessian inverse to mimic the behavior of Newton's algorithm. Formally, gradient descent with preconditioning is defined as:

$$W_{\ell_{t+1}} = W_{\ell_t} - \eta_{t+1} \, P_{t+1}^{-1} \, g_{\ell_{t+1}},$$ (52)

where *preconditioner* $P_{t+1}$ is often a positive-definite matrix. A critical interpretation of preconditioner is their role in performing gradient descent in a transformed coordinate system, which can be viewed as a mapping from gradients to that system of interest. Accordingly, we reformulate and interpret the preconditioner in Equation 52 as an associative memory that maps the set of gradients (or a function of gradients denoted as $g$) to the system of our choice, denoted as $\hat{g}$:

$$W_{\ell_{t+1}} = W_{\ell_t} - \eta_{t+1} \, P_{t+1}^{-1}\left(g_{\ell_{t+1}}\right),$$ (53)

where internally (in a nested level), $P_{t+1}$ learns how to perform this mapping using an objective:

$$\min_P \quad \tilde{\mathcal{L}}\left(P\left(\hat{g}\right); g\right).$$ (54)

Given this viewpoint, the main question is about finding the best coordinate system that can empower the compression process. The most simple variant is an identity mapping, where we preserve the metric system and use $P$ to map $g$ (i.e., gradients in this case) to itself, resulting in preconditioning terms in Adam [33] and AdaGrad [108]. These results, along with the representation of Adam and its variants as associative memories, show that not only momentum-based optimizers are associative memories, but they also can be decomposed into a set of nested learning problems, each of which optimized with gradient descent. In a more general form, however, one can use more nested levels and optimize the inner problems in Equation 54 with gradient descent, resulting in:

$$P_{t+1} = P_{t+1} - \zeta_{t+1} \nabla_{P_t} \tilde{\mathcal{L}}\left(P_t; g_{t+1}, \hat{g}_{t+1}\right).$$ (55)

In the NL framework, to design an effective preconditioning, one needs to find the right choice of $\hat{g}$ and $\tilde{\mathcal{L}}$. This viewpoint can also lead to other classes of algorithms with gradient/momentum orthogonalization: e.g., Muon and its variants [34]. Recalling Muon optimizer [34]:

$$W_{\ell_{t+1}} = W_{\ell_t} + \texttt{NewtonSchulz}_k\left(m_{\ell_{t+1}}\right)$$
$$m_{\ell_{t+1}} = \alpha_{\ell,t+1}m_{\ell_t} - \eta_{\ell,t+1}\nabla_{W_{\ell_t}}\mathcal{L}\left(W_{\ell_t};x_{t+1}\right), \tag{56}$$

where $\texttt{NewtonSchulz}_k(\cdot)$ performs $k$ steps of Newton-Schulz orthogonalization process. From the above discussion about the general formulation of preconditioning, one can see $\texttt{NewtonSchulz}_k(\cdot)$ operator as a mapping from gradients of momentum term to a proper metric system. The choice of proper coordinate system in Muon is to orthogonalize the gradients and so we aim to find a mapping $P\left(\cdot\right)$ by minimizing a loss function $\min_P \tilde{\mathcal{L}}(P;O,m)$ where objective $\tilde{\mathcal{L}}(\cdot;\cdot,\cdot)$ measures the quality of mapping from $O$ to either $m$ or $g$ by $P(\cdot)$. A critical challenge in this process is that the parameter $O$ itself is not given and so the mapping requires learning both the mapping and the proper orthogonal space. A simple formulation measuring orthogonalization, can be achieved by defining the objective as:

$$\tilde{\mathcal{L}}(P(g);g) = \left\| P(g)^\top P(g) - I \right\|_F^2, \tag{57}$$

where $P(g)$ is the orthogonal space that we aim to directly learn from gradients. This objective ensures that the gradients (or momentum) and their mapping are relatively close while the mapping is to an orthogonal space. Optimizing the above objective to find $O = P(g)$ with one step of gradient descent results in:

$$O_{i+1} = O_i - \zeta_{i+1}\nabla_{O_i}\tilde{\mathcal{L}}\left(Oi;g_t\right) = O_i - \zeta_{i+1}\left(O_i - g_t + 2O_i\left(O_i^\top O_i - I\right)\right), \tag{58}$$

which recovers the 3-degree polynomial (initial value $O_0 = g_t$). In a summary, the higher-frequency level learns the orthogonal mapping and then the lower-frequency process use the learned mapping to optimize the weights.

# E  Details on Hope Architecture

A general formulation for the associative memory-based blocks is to project the data into keys, values, and queries and learns how to map keys to values and how to retrieve from the mapping based on queries. More formally, for a parametric associative memory, let $x_t \in \mathbb{R}^d$ for $t = 1, \ldots, L$ be the input, we have:

$$k_t = x_t W_k, \qquad v_t = x_t W_v, \qquad q_t = x_t W_q, \qquad \eta_t = x_t W_\eta, \qquad \alpha_t = x_t W_\alpha, \tag{59}$$
$$\min_{\mathcal{M}} \ \mathcal{L}\left(\mathcal{M};k_t,v_t\right), \qquad \text{with an optimization algorithm} \tag{60}$$
$$y_t = \mathcal{M}_t q_t. \tag{61}$$

For the sake of clarity, we use red (resp. blue) to highlight computations/weight in the upper level (resp. lower level). We can add a new level for each of $W_k, W_v, W_q, W_\eta$, and $W_\alpha$ and allow them to be updated in-context. For the sake of efficiency, a simple version is to share the values for all the components in the nested system of associative memories:

$$k_t = \mathcal{M}_{k,t-1}\left(x_t\right), \qquad v_t = \mathcal{M}_{v,t-1}\left(x_t\right), \qquad q_t = \mathcal{M}_{q,t-1}\left(x_t\right), \qquad \eta_t = \mathcal{M}_{\eta,t-1}\left(x_t\right),$$
$$\alpha_t = \mathcal{M}_{\alpha,t-1}\left(x_t\right), \tag{62}$$
$$\min_{\mathcal{M}_\square} \ \mathcal{L}\left(\mathcal{M}_\square;\square_t,v_t\right), \qquad \text{with an optimization algorithm,} \quad \square \in \{k,v,q,\eta,\alpha\}, \tag{63}$$
$$\min_{\mathcal{M}_{\text{mem}}} \ \mathcal{L}\left(\mathcal{M}_{\text{mem}};k_t,v_t\right), \qquad \text{with an optimization algorithm,} \tag{64}$$
$$y_t = \mathcal{M}_{\text{mem},t}\left(q_t\right), \tag{65}$$

where the initial states of all memories, i.e., $\mathcal{M}_{\square,0}$ for any $\square \in \{k,v,q,\eta,\alpha,\text{memory}\}$ are meta-learned across all sequences/contexts. As discussed earlier, the meta-learning of the initial states of memories is essential for both fast-adaption, training stability, robustness to noise in the data.

This design provides a fully adaptive memory, where all the components can adapt themselves in-context. It, however, (1) still lacks self-modification, where the model in response to new data

changes its own parameters or learning process [115]; (2) has suboptimal design as it shares of keys and values for all the memories. In continual learning, where the model requires consistent weight/knowledge update in response to new data, it is critical for the model to not solely rely on data, and instead learns how to modify itself when it is needed. Motivated by the above points, and inspired by the self-modifying mechanisms that generate their own values based on the context [94, 115, 116], we present self-modifying deep associative memory where the models generate their own values:

$$\boldsymbol{y}_t = \mathcal{M}_{\text{memory},t-1}\left(\boldsymbol{q}_t\right), \qquad \boldsymbol{k}_t = \mathcal{M}_{\boldsymbol{k},t-1}\left(\boldsymbol{x}_t\right), \qquad \boldsymbol{v}_t = \mathcal{M}_{\boldsymbol{v},t-1}\left(\boldsymbol{x}_t\right), \qquad \eta_t = \mathcal{M}_{\eta,t-1}\left(\boldsymbol{x}_t\right),$$
$$\alpha_t = \mathcal{M}_{\alpha,t-1}\left(\boldsymbol{x}_t\right), \tag{66}$$
$$\hat{\boldsymbol{v}}_{\square,t} = \mathcal{M}_{\square,t-1}\left(\boldsymbol{v}_t\right), \qquad\qquad \text{(Generating its own values for each memory)} \tag{67}$$
$$\min_{\mathcal{M}_{\square}}\ \mathcal{L}\left(\mathcal{M}_{\square};\boldsymbol{k}_t,\hat{\boldsymbol{v}}_{\square,t}\right), \qquad \text{with an optimization algorithm,} \quad \square \in \{\boldsymbol{k},\boldsymbol{v},\boldsymbol{q},\eta,\alpha,\text{memory}\},$$
$$\tag{68}$$

where $\boldsymbol{q}_t = \boldsymbol{x}_t W_q$ is the only non-adaptive projection, $\eta_t$ is the learning rate in optimization process, and $\alpha_t$ is the retention gate (forget gate or weight decay) in the optimization process. Note that, again, the initial states of all memories, i.e., $\mathcal{M}_{\square,0}$ for any $\square \in \{\boldsymbol{k},\boldsymbol{v},\boldsymbol{q},\eta,\alpha,\text{memory}\}$ are meta-learned across all sequences/contexts, and so are optimized in the higher levels (or outer-loop).

Learning the mappings for associative memory modules (see Equation 68) requires a choice of optimization algorithm as well as an objective $\mathcal{L}$ that measures the quality of mappings. A simple and common choice for objective and optimization process are $L_2$-regression loss, and gradient descent algorithm. As for the objective, we use $L_2$-regression loss, i.e., $\mathcal{L}(\mathcal{M};\boldsymbol{k},\boldsymbol{v}) = \|\mathcal{M}(\boldsymbol{k}) - \boldsymbol{v}\|_2^2$. As discussed earlier, the choice of optimizer highly depends on the context of optimization. For example, gradient descent from associative memory perspective is based on dot-product similarity and so the update at each step, is solely based on the input and does not incorporate the previous data samples to the update. When performing optimization in the token space, however, we know tokens are highly correlated. Therefore, we use our DGD with weight decay, resulting in general update rule of:

$$\boldsymbol{y}_t = \mathcal{M}_{\text{memory},t-1}\left(\boldsymbol{q}_t\right), \qquad \boldsymbol{k}_t = \mathcal{M}_{\boldsymbol{k},t-1}\left(\boldsymbol{x}_t\right), \qquad \boldsymbol{v}_t = \mathcal{M}_{\boldsymbol{v},t-1}\left(\boldsymbol{x}_t\right), \qquad \eta_t = \mathcal{M}_{\eta,t-1}\left(\boldsymbol{x}_t\right),$$
$$\alpha_t = \mathcal{M}_{\alpha,t-1}\left(\boldsymbol{x}_t\right), \tag{69}$$
$$\hat{\boldsymbol{v}}_{\square,t} = \mathcal{M}_{\square,t-1}\left(\boldsymbol{v}_t\right), \qquad\qquad \text{(Generating its own values for each memory)}$$
$$\tag{70}$$
$$\mathcal{M}_{\square,t} = \mathcal{M}_{\square,t-1}\left(\alpha_t \boldsymbol{I} \;-\; \eta_t \boldsymbol{k}_t \boldsymbol{k}_t^\top\right) - \eta_t \nabla \mathcal{L}_{\mathcal{M}_{\square,t-1}}\left(\mathcal{M}_{\square,t-1};\boldsymbol{k}_t,\hat{\boldsymbol{v}}_{\square,t}\right), \tag{71}$$
$$\square \in \{\boldsymbol{k},\boldsymbol{v},\boldsymbol{q},\eta,\alpha,\text{memory}\}.$$

Here, the architecture of the memories are arbitrary and even we are not forced to use the same architecture for all components. We use a 2-layer MLP block as the architecture of all the memories:

$$\mathcal{M}_{\square}(\cdot) = (\cdot) + W_{\square,1}\sigma(W_{\square,2}(\cdot)). \tag{72}$$

### E.1 Fast and Parallelizable Training

In the above, we discussed how to design a model that can learn to generate its own latent values and so modify itself. The main challenge from the practical point of view is the efficiency of the method and if its training is parallelizable. We follow the chunk-wise training algorithm of non-linear update rules [28, 65] and use update frequency of $f_{\square} = \frac{L}{C_{\square}}$, where $L$ is the context length. While there is no limitation to use different chunk-sizes, in our experiments, we use two different value of chunk sizes, one for the update of $\mathcal{M}_{\text{memory}}(\cdot)$ and the other for all the other memories in the self-referential Titans.

In more details, given an input sequence $\{\boldsymbol{x}_t\}_{t=1}^L$ and chunk size $1 \leq C \leq L$, we split the sequence into $\lceil \frac{L}{C} \rceil$ chunks of $\{\boldsymbol{x}_{((i-1)C+t)}\}_{t=1}^C$ for $i = 1,\ldots,\lceil \frac{L}{C} \rceil$, and then generate all elements in Equation 69 at the end of each chunk for the next chunk. This allows for generating all the elements for the entire chunk in parallel, before starting the computation for this chunk. Furthermore, to update the memory modules based on Equation 71, we take the gradient with respect to the last state of the previous chunk. Again, this allows for computing all the gradients for the next chunk in parallel. In more details, given this chunk-wise updating procedure, the update rule for the self-referential Titans is computed as:

$$\boldsymbol{y}_t = \mathcal{M}_{\text{memory},C \times \lceil \frac{t}{C} \rceil}(\boldsymbol{q}_t), \quad \boldsymbol{k}_t = \mathcal{M}_{\boldsymbol{k},C \times \lceil \frac{t}{C} \rceil}(\boldsymbol{x}_t), \quad \boldsymbol{v}_t = \mathcal{M}_{\boldsymbol{v},C \times \lceil \frac{t}{C} \rceil}(\boldsymbol{x}_t),$$

$$\eta_t = \mathcal{M}_{\eta,C \times \lceil \frac{t}{C} \rceil}(\boldsymbol{x}_t), \quad \alpha_t = \mathcal{M}_{\alpha,C \times \lceil \frac{t}{C} \rceil}(\boldsymbol{x}_t), \tag{73}$$

$$\hat{\boldsymbol{v}}_{\square,t} = \mathcal{M}_{\square,C \times \lceil \frac{t}{C} \rceil}(\boldsymbol{v}_t), \qquad\qquad \text{(Generating its own values for each memory)}$$

$$\mathcal{M}_{\square,t} = \mathcal{M}_{\square,t-1}\left(\alpha_t \boldsymbol{I} - \eta_t \boldsymbol{k}_t \boldsymbol{k}_t^\top\right) - \eta_t \nabla \mathcal{L}_{\mathcal{M}_{\square,C \times \lceil \frac{t}{C} \rceil}}\left(\mathcal{M}_{\square,C \times \lceil \frac{t}{C} \rceil}; \boldsymbol{k}_t, \hat{\boldsymbol{v}}_{\square,t}\right), \tag{74}$$

$$\square \in \{\boldsymbol{k}, \boldsymbol{v}, \boldsymbol{q}, \eta, \alpha, \text{memory}\}.$$

Here, the architecture of the memories are arbitrary and even we are not forced to use the same architecture for all components. We use a 2-layer MLP block as the architecture of all the memories:

$$\mathcal{M}_\square(\cdot) = (\cdot) + W_{\square,1}\sigma(W_{\square,2}(\cdot)). \tag{75}$$

Since all the gradients as well as new keys, values, learning-rates, and weight decays can be computed in parallel before starting the processing of the current chunk, the above updates accepts the fast parallelizable dual form that is discussed by Sun et al. [65] and Behrouz et al. [28]. To better illustrate the above update rule for self-referential Titans, let us derive the recurrent formula for the simplest case of matrix-valued memory. We derive the recurrent form for two different objectives:

- Dot-product similarity $\mathcal{L}(\mathcal{M}; \boldsymbol{k}, \boldsymbol{v}) = -\langle \mathcal{M}\boldsymbol{k}, \boldsymbol{v} \rangle$: Given this objective and linear memory, the gradient is calculated as $\boldsymbol{v}\boldsymbol{k}^\top$, which results in update rule of:

$$\mathcal{M}_{\square,t} = \mathcal{M}_{\square,t-1}\left(\alpha_t \boldsymbol{I} - \eta_t \boldsymbol{k}_t \boldsymbol{k}_t^\top\right) - \eta_t \hat{\boldsymbol{v}}_{\square,t} \boldsymbol{k}_t^\top, \tag{76}$$

$$\square \in \{\boldsymbol{k}, \boldsymbol{v}, \boldsymbol{q}, \eta, \alpha, \text{memory}\}$$

- $L_2$-regression loss: Given this objective and linear memory, the gradient is calculated as $(\mathcal{M}\boldsymbol{k} - \boldsymbol{v})\boldsymbol{k}^\top$, which results in update rule of:

$$\mathcal{M}_{\square,t} = \mathcal{M}_{\square,t-1}\left(\alpha_t \boldsymbol{I} - \eta_t \boldsymbol{k}_t \boldsymbol{k}_t^\top\right) - \eta_t \left(\mathcal{M}_{\square,C \times \lceil \frac{t}{C} \rceil} \boldsymbol{k}_t - \hat{\boldsymbol{v}}_{\square,t}\right) \boldsymbol{k}_t^\top, \tag{77}$$

$$\square \in \{\boldsymbol{k}, \boldsymbol{v}, \boldsymbol{q}, \eta, \alpha, \text{memory}\}.$$

### E.2 Hope Neural Learning Module

In the previous sections, we first discussed Continuum Memory System (CMS) that allows for more persistent storage of memories and defines memory as a spectrum of blocks with different frequencies of update. Due to the larger capacity and constraints for scaling the parameters, often CMS requires simple learning rule but higher capacity to store more persistent knowledge. On the other hand, in the previous section, we discussed the design of a self-modifying Titans, where it can generate its own keys and so learning update to better adapt to the context. Contrary to CMS, the self-modifying Titans has a small capacity but is using a complex and expressive learning rule. Accordingly, these two systems seem to be complementary and their combination can enhance the model expressiveness from different aspects.

To this end, we present HOPE architecture: A neural learning module that incorporates self-modifying Titans followed by Continuum Memory System. The HOPE design is illustrated in Figure 3. Formally, let $\boldsymbol{x}_t \in \mathbb{R}^d$ for $t = 1, \ldots, L$ be the input, the HOPE forward pass is defined as (we remove the normalization and convolution layers for the sake of clarity):

$$\boldsymbol{o}_t = \mathcal{M}_{\text{memory},t-1}(\boldsymbol{q}_t), \qquad \boldsymbol{k}_t = \mathcal{M}_{\boldsymbol{k},t-1}(\boldsymbol{x}_t), \qquad \boldsymbol{v}_t = \mathcal{M}_{\boldsymbol{v},t-1}(\boldsymbol{x}_t), \tag{78}$$

$$\eta_t = \mathcal{M}_{\eta,t-1}(\boldsymbol{x}_t), \qquad \alpha_t = \mathcal{M}_{\alpha,t-1}(\boldsymbol{x}_t), \tag{79}$$

$$\hat{\boldsymbol{v}}_{\square,t} = \mathcal{M}_{\square,t-1}(\boldsymbol{v}_t), \tag{80}$$

$$\mathcal{M}_{\square,t} = \mathcal{M}_{\square,t-1}\left(\alpha_t \boldsymbol{I} - \eta_t \boldsymbol{k}_t \boldsymbol{k}_t^\top\right) - \eta_t \nabla \mathcal{L}_{\mathcal{M}_{\square,t-1}}\left(\mathcal{M}_{\square,t-1}; \boldsymbol{k}_t, \hat{\boldsymbol{v}}_{\square,t}\right), \tag{81}$$

$$\square \in \{\boldsymbol{k}, \boldsymbol{v}, \boldsymbol{q}, \eta, \alpha, \text{memory}\}.$$

$$\boldsymbol{y}_t = \text{MLP}^{(f_k)}(\text{MLP}^{(f_{k-1})}(\cdots \text{MLP}^{(f_1)}(\boldsymbol{o}_t))), \tag{82}$$

where the block's output for token $t$ is $\boldsymbol{y}_t$. In our experiments, we also normalize $\boldsymbol{q}$ and $\boldsymbol{k}$ with $L_2$ normalization and also use local convolutions with window size of 4.

**Hope-Attention.** We also use another variant of HOPE, in which we simply replace the self-modifying Titans with softmax global attention [27].

Table 2: Architectural Details.

| Model | Block | Dim | Head | Peak LR | Token |
|-------|-------|------|------|---------|-------|
| 170M  | 12    | 768  | 16   | 3e-3    | 15B   |
| 340M  | 24    | 1024 | 16   | 1.5e-3  | 15B   |
| 760M  | 24    | 1536 | 16   | 1.25e-3 | 30B   |
| 1.3B  | 18    | 2048 | 8    | 7e-4    | 100B  |

## F   Experimental Setups

To study HOPE as a backbone of a language model and evaluate it on common language modeling and common-sense reasoning tasks with the setup of:

- Datasets: We evaluate HOPE and baselines on Wikitext [69], LMB [70], PIQA [71], HellaSwag [72], WinoGrande [73], ARC-easy (ARC-e) and ARC-challenge (ARC-c) [74], SIQA [75], and BoolQ [117] benchmarks.

- Baselines: As for the baselines, we use RetNet [61] and DeltaNet [63] as the representatives of the models that are *purely* based on Hebbian- or Delta-rule, and two modern *matrix-valued* recurrent models with the best performance compared to others: i.e., RWKV-7 [118] and Comba [67]. As another group of baselines, we compare with attention-free *deep* memory modules with diverse internal attentional bias of dot-product, $L_2$, and $L_p$ regression: i.e., TTT [65], Miras [58], DLA [68] and Titans [28]. Finally, we also compare with Transformers [27] as well as the hybrid of attention and linear RNN, Samba [119].

- Training: We train models with about 760M and 1.3B parameters, trained with 30B and 100B tokens, respectively, from a mixture of FineWeb-Edu [120] and long-context documents with a vocabulary size of 32K to train all the models from scratch. All models are trained with standard next-token prediction for language modeling, optimized using AdamW with tuned learning rate for each model, and with the default optimizer configuration as in Behrouz et al. [28].

