# OpenReview forum: "Nested Learning: The Illusion of Deep Learning Architectures"
_NeurIPS.cc/2025/Conference — NeurIPS 2025 poster_

### Official Review · Reviewer_s9eJ · 2025-06-29

**Clarity:** 3
**Significance:** 3
**Originality:** 3
**Rating:** 4
**Confidence:** 3

**Summary:**

This paper introduces the Nested Learning paradigm, a theoretical framework that reframes deep learning models and their training procedures as a unified system of nested optimization problems. The authors propose that the conventional separation between neural architecture design and optimizer choice can be decomposed into a hierarchy of simpler optimization tasks. The authors posit that NL provides a general language that encompasses and extends existing concepts, including meta-learning and in-context learning.

This framework leads to a proposed family of general optimizers with high expressivity. Building upon these insights, the authors present a self-referential memory model, HOPE, designed to be an in-context learner at multiple levels of abstraction. The paper shows that HOPE achieves competitive performance on language modeling, common-sense reasoning, and long-context recall tasks, when compared to Transformers and Mamba.

**Questions:**

Please see above

**Ethical Concerns:**

["NO or VERY MINOR ethics concerns only"]

**Final Justification:**

The authors addressed my concerns about writing and experimental results.

**Limitations:**

Yes

**Paper Formatting Concerns:**

No.

**Quality:**

3

**Strengths And Weaknesses:**

Strengths:
1. The idea to formulate ICL as an optimization algorithm brings new insights into how to conceptually understand the intuition behind ICL.
2. Hope demonstrates good performance at a similar training scale.

Weakness:
1. The author should provide a list (even if not exhaustive) of what can be defined as a prime neural architecture. This is the cornerstone of the author's formulation; however, it is difficult to understand what can be considered as such a cornerstone for building a neural network per the definition.
2. Similarly, it's not entirely clear how Line 192 eliminates the need to backpropagate between building blocks.
3. The avg of HOPE is 0 in Table 1. In general, the writing needs a lot more polishing.

---

> ### Author Rebuttal · Authors · 2025-07-31
>
> We thank the reviewer for their time and valuable comments. We are glad that you found our work with new insights. We respond to your comments below:
>
> > The author should provide a list (even if not exhaustive) of what can be defined as a prime neural architecture. This is the cornerstone of the author's formulation; however, it is difficult to understand what can be considered as such a cornerstone for building a neural network per the definition.
>
> **Response**: Great suggestion!. We provide a list of such architectures as follows:
> **Prime neural Modules** (or non-decomposable modules) $\rightarrow$
> - Examples are: MLP, Gated MLP, Convolutions, Gated Convolutions, Multi-resolution Convolutions, any combination of MLPs that only fuse the information across features (not sequence), etc.
>
> **Non-Primal Modules**  $\rightarrow$
> -  Are modules that can be decomposed into nested blocks such as linear attention: $M_t = M_{t-1} + v_t k_t^{T}$ can be decomposed into a new level, in which given $L_t = <M_{t} k_t, v_t>$ we optimize this loss function using gradient descent in the inner-loop:
> $M_t = M_{t-1} - \nabla L_t  \rightarrow$ since $\nabla L_t = - v_t k_t^{T}$ we have: $M_t = M_{t-1} + v_t k^{T}_t.$
>
> Therefore, an example for non-primal modules is linear attention, or any other recurrent architecture such as DeltaNet, Titans, etc.
>
> We will make sure to incorporate these examples in the final version of the paper to make the concept of prime neural network more clearer.
>
>
> > Similarly, it's not entirely clear how Line 192 eliminates the need to backpropagate between building blocks.
>
> **Response**:  There seems to be a misunderstanding here. Please note that if there is a backpropagation between two building blocks, then they have been in the same block in the first place as they share the gradient flow. However, as we discussed in the paper, different levels of a learning module might not share the gradient flow as the nature of their context can be different. For example, the context for the first level is the gradient of parameters, while the context for the internal gradient flow of a recurrent model is the tokens in the input sequence.
>
> In this regard, please see our illustrative example in Sec. 2.2. The inner-loop of the recurrent model has its own gradient flow as the gradient is with respect to the tokens and the recurrence is over the sequence length. On the other hand, the outer-loop aims to optimize the parameters in the pre-training by using gradient descent on the entire data points. Therefore, the recurrence is over the data samples and is not on the tokens any more.
>
> As another example, considering momentum term as another level, it aims to compress the gradient terms, and so its iteration is over gradients.
>
> All in all, we do not force the model to eliminate the need to backpropagate between building blocks.  It is indeed a fact about how neural networks are decomposed into these building blocks.
>
> We hope that this has clarified the misunderstanding. Please let us know if further clarification is needed and we would be happy to provide more examples/clarifications.
>
> > The avg of HOPE is 0 in Table 1. In general, the writing needs a lot more polishing.
>
> Thank you very much for bringing this to our attention. We will make sure to fix this issue in the final version. We have provided the average in the following table:
>
> | Model Size    | Average |
> | -------- | ------- |
> | 340 M |  46.90   |
> | 760 M |   52.25   |
> | 1.3 B  |  57.23   |
>
>
>
> > General Writing:
>
> Regarding the general writing we will extensively proofread the paper and clarify the points that are unclear. Also, if there is a specific part that you would like  us to clarify further, please let us know and we would be happy to provide additional details and improve it.
>
> Please also note that, since this study is advocating for a new learning paradigm, it is expected and possible that several terms and concepts are new and might not be fully familiar to what we often read in deep learning literature. This would also make the paper more challenging to read and comprehend, which comes from the nature of our work. We kindly ask you to consider this point that this is the initial work in this direction. We will do our best to improve the writing in the final version of the paper and make it easy to follow.
>
>
> ---
> ## General Note on Contributions:
> ---
>
> We want to kindly bring some points about the contributions of our work to your consideration: (1) In our submission, we not only present a new learning paradigm that can explain and unifies several well-known concepts, but we also present a new architecture that provides a noticeable performance benefit over the baselines in diverse benchmarks. In addition, we study and compare the in-context learning ability of different architectures in the Appendix, which alone has often been the only focus of the similar studies.  (2) While the current focus of the community is  on improving scaling of the architectures parameters and data, Nested Learning presents a new dimension to enhance the model capability, which is its internal computations. We believe this paradigm can help to design more efficient architectures with fewer parameters, and inspire more research studies to build on .  (3) Due to the recent community interest in designing alternative sequence models and understanding the in-context learning ability of  different architectures, we believe that Nested Learning is a timely contribution and its publication can help the community to better design architectures capable of in-context learning and better design architecture-specific optimizers that can lead to better performance, even for current models.
>
>
>
> **We hope that our responses have addressed your concerns. Please let us know if there are additional concerns and we would be more than happy to engage more and clarify any remaining concerns.**

---

> > ### Comment · Reviewer_s9eJ · 2025-08-06
> >
> > Thank you for your clarifications. I will increase my rating to 4.

---

> ### Author Response · Authors · 2025-08-05
>
> Dear Reviewer,
>
> Thank you for your time and review! We sincerely thank you for your time, effort, and thoughtful review and for recognizing the new insights presented by our work. We hope our responses have fully addressed your concerns. We would be very happy to continue the discussion and answer any remaining questions or concerns. If you find our response satisfactory, we would be grateful if you would consider raising your score.

---

### Official Review · Reviewer_P97x · 2025-07-01

**Clarity:** 3
**Significance:** 3
**Originality:** 3
**Rating:** 4
**Confidence:** 3

**Summary:**

Neural network architectures and optimization algorithms are traditionally treated as separate design concerns, but The authors propose Nested Learning (NL) paradigm treats them as integrated layers of nested optimization problems. they formalize NL, show how it subsumes meta-learning and mesa-learning, introduce a new optimizer family, and propose a self-referential model called HOPE that adapts internal memory and projections at multiple levels.

**Questions:**

See the weakness section above.

extra questions:

1- How scalable is NL in practice? Does adding more nested levels incur significant computational overhead or instability?

2- How do you handle nested levels in backpropagation?

3- Table 4 in appendix H, show that HOPE is better at memorization and selective copying rather than ICR or noisy ICR. How do you explain that?

4- How does HOPE perform with extremely long context datasets? e.g., BABILong

**Ethical Concerns:**

["NO or VERY MINOR ethics concerns only"]

**Final Justification:**

the approach is novel and the empirical results are impressive. But the paper is not well-written, i.e., some concepts are not well explained etc. The authors promised to polish the writing in the final version. Hence, I am more inclined to accept. BUt I don't have a strong opinion.

**Limitations:**

Yes

**Quality:**

3

**Strengths And Weaknesses:**

**Strengths:**

- Original Perspective: Reframing architectures and optimizers under a unified Nested Learning framework is somehow a novel idea, particularly if it can subsume both meta-learning and in-context learning (ICL).

- Technical Depth:  I did not go through all the proofs in details but the theoretical grounding is rigorous. for example, studying momentum and Muon as associative memory-based NL instances is interesting.

- Concrete Contributions: based on the proposed formulation of Nested Learning and the insights gained, the authors propose a new model, called HOPE, which a self-referential deep memory model. The proposed approach achieves strong results compared to competing methods.


**Weaknesses:**

- There is a heavy reliance on terms like “prime neural modules”, “value-less associative memory”, “non-decomposable modules”, etc., without sufficiently grounding them in intuition or concrete examples. For example the definition of 'Prime Neural Module' provided in L182 is not clear. can the authors provide a concrete examples of primal vs non-primal?

- Even though, the nested optimization formulation is mathematically described, the distinction between levels, gradient flows, and objectives is sometimes vague.

> Each problem has its own gradient flow and there is no backpropagation between different levels and blocks.

This is both a strong assumption and an architectural decision. It needs clearer justification and implications.

- Reproducibility issues: the authors don't share any codes that can be used to reproduce their results.

**Minor errors:**

L17 “further enhancing the interpretability, and explain how...” → fix agreement (“explaining” or split sentence)\

L106 “in which a neural architecture parameterized by θ is trained...” → repetition and long clauses
You use

*Notation: Ensure notation is consistent.*
- For example, In L145, you define $L^{(1)}$ as the next token prediction loss but in L157, you use   $L^{(2)}$ to refer the it.

- Another example, You switch between using $l$, $L$, and $\mathcal{L}$ to refer to loss/objective functions. For instance: $l$ is used both in Equation (1) as  $l(\theta,\mathcal{T}\_i,\Phi)$  and Equation (2) as $l(f(x_i), \mathcal{M}_{\theta}(x_1,f(x_1),...,x_i))$. This is inconsistent. can you make it rigorous and use different notations to define different losses.

- You use $M_k$ for key modules but you also define W_k. this leads to confusion. Is M_k a memory or projection matrix? same goes for $M_q$ and $M_v$

---

> ### Author Rebuttal · Authors · 2025-07-31
>
> We thank the reviewer for their time and valuable comments. We are glad that you found our framework a novel idea. We respond to your comments below:
>
> >  There is a heavy reliance on terms like “prime neural modules”, “value-less associative memory”, “non-decomposable modules”, etc., without sufficiently grounding them in intuition or concrete examples. For example the definition of 'Prime Neural Module' provided in L182 is not clear. can the authors provide a concrete examples of primal vs non-primal? ...
>
> **Response**: Thank you for bringing this to our attention. We understand the importance of providing examples for these new concepts and following your suggestions, we will make sure to provide additional examples and a table to summarize examples for each of our new terms.
>
> More specifically, we will include a more comprehensive version of the following examples:
>
> **Prime neural Modules** (or non-decomposable modules) $\rightarrow$
> - Examples are: MLP, Gated MLP, Convolutions, Gated Convolutions, Multi-resolution Convolutions, any combination of MLPs that only fuse the information across features (not sequence), etc.
>
> **Non-Primal Modules**  $\rightarrow$
> -  Are modules that can be decomposed into nested blocks such as linear attention: $M_t = M_{t-1} + v_t k_t^{T}$ can be decomposed into a new level, in which given $L_t = <M_{t} k_t, v_t>$ we optimize this loss function using gradient descent in the inner-loop:
> $M_t = M_{t-1} - \nabla L_t  \rightarrow$ since $\nabla L_t = - v_t k_t^{T}$ we have: $M_t = M_{t-1} + v_t k^{T}_t.$
>
> Therefore, an example for non-primal modules is linear attention, or any other recurrent architecture such as DeltaNet, Titans, etc.
>
> **Value-less Associative Memory** $\rightarrow$
> - Associative memory that maps all keys to a single value, such as recurrent models with formulation: $h_{t} = A h_{t-1} + f(k_t)$, where $f(.)$ is an arbitrary function of input keys $k_t$.
>
>
> We will further clarify these concepts:
>
> - **Gradient Flow**: Each box (in Figure 1), has its own gradient flow. For example, in the above example of linear attention, we can see that optimizing the objective $L_t = <M_{t} k_t, v_t>$ using gradient descent is across sequence length and is different from the gradient flow of optimizing the entire parameters of the model (i.e., during pre-training phase). Therefore, we say there are some boxes (or sub-modules), each of which with its own gradient flow.
> - **Objectives**: Please note that the optimization process in each sub-module requires an objective to be optimized. Therefore, we refer to the loss function of each sub-module as its objective. For example, in the above example, $L_t = <M_{t} k_t, v_t>$ is the objective of that sub-module.
> - **Levels**: Each of sub-modules with their own gradient flow is inside a larger optimization problem. For example, as discussed above, the optimization of objective $L_t = <M_{t} k_t, v_t>$, which is equivalent to linear attention, is inside a larger optimization problem, in which we optimize the parameters of the entire network (for example MLP layers, normalizations, projections, embeddings, etc.). We define a partial order on these problems and so refer to each position in this ordered list as a level. For example, the largest optimization process is the optimization of the parameters of the entire neural network. This optimization will be in level one. Next, its direct inner-optimization problem is the optimizing process of $L_t = <M_{t} k_t, v_t>$, and so this will be in level 2, and so on and so forth.
>
> Following your suggestion, we will make sure to provide additional explanations and examples to further clarify these concepts.
>
>
> > “Each problem has its own gradient flow and there is no backpropagation between different levels and blocks”. This is both a strong assumption and an architectural decision. It needs clearer justification and implications.
>
> **Response:** Please note that this has not been an assumption or architectural decision but part of the modeling paradigm (which in fact accurately models the internal process of the architectures). Please note that if there is a backpropagation between different levels and blocks, then they are sharing a gradient flow and so has been one block from the beginning.
>
> To further clarify: Please note that for each sub-module (box in Figure 1), the other parameters outside of it are considered hyperparameters and so backpropagation for each sub-module has its own path. Let’s consider a special case, where we have a fully nested problem, with only two levels. This case is equivalent to meta-learning, in which the parameters of the inner-loop are considered hyperparameters for the out-loop optimization and vice versa. Therefore, the outer-loop has its own backpropagation and so does not go through the parameters of the inner-loop. The same is also valid for the inner-loop.
>
> The above example, while illustrating a special case, is valid for all NL instances.
>
>
>
> > Reproducibility issues: the authors don't share any codes that can be used to reproduce their results.
>
> **Response**: We thank the reviewer and we understand the importance of open science. We kindly want to bring to your consideration that sometimes sharing the code is not possible due to circumstances that are not even in the control of authors. However, following your suggestion, we adhere and will make sure to provide all the details that are needed for clarifying the experimental setup and reproducibility of the results.
>
> > Minor errors
>
> **Response**: Thank you very much for bringing them to our attention. We will make sure to proofread the paper and fix all typos. Also, you are right and we will make sure to fix this inconsistent use of notations in the final version of the paper.
>
>
> > How scalable is NL in practice? Does adding more nested levels incur significant computational overhead or instability?
>
> **Response**: Please note that we have scaled our results to 1.3B, which is considered relatively large for academic papers and the resulting model did not show instability. However, we expect that excessively increasing the number of levels can cause instability in the training, which is an important direction for future study to improve NL-based models. We will make sure to discuss these limitations in the paper.
>
> > How do you handle nested levels in backpropagation?
>
> **Response**: Please note that for each sub-module (box in Figure 1), the other parameters outside of it are considered hyperparameters and so backpropagation for each sub-module has its own path. For further clarifying this, let’s consider a special case, where we have a fully nested problem, with only two levels. This case is equivalent to meta-learning, in which the parameters of the inner-loop are considered hyperparameters for the out-loop optimization and vice versa. Therefore, the outer-loop has its own backpropagation and so does not go through the parameters of the inner-loop. The same is also valid for the inner-loop.
> The above example, while illustrating a special case, is valid for all NL instances.
>
> > Table 4 in appendix H, shows that HOPE is better at memorization and selective copying rather than ICR or noisy ICR. How do you explain that?
>
> **Response:** Thank you for mentioning that. Please note that Hope achieves the best performance across all tasks, except one of them which is fuzzy ICR. While we can compare the models based on the average of their performance across tasks, there is always a possibility that for some individual tasks, mainly because of a specific data type or distribution, a model underperforms a baseline. We believe for this specific case the more complexity of Hope can make it more data hungry than a simpler model like Titans. Since the training is small in this benchmark, for this specific task, we can see a small performance difference.
>
> > How does HOPE perform with extremely long context datasets? e.g., BABILong
>
> **Response:** Please note that the original setup in BABILong benchmark is based on the fine-tuning models and so benchmarks like RULER are more commonly used in the similar study to show the performance of models in longer contexts. Also, in our RULER experiments, we observe that even in this simple setup, most baselines achieve near zero accuracy and so the advantage of Hope over other recurrent models are already clear.
>
> However, following your suggestion, based on the BABILong benchmark setup, we are fine-tuning our models and will report the results when they are ready. We also are adhered to provide these results in the final version of the paper.
>
>
>
>
> **We hope that our above responses have addressed your concerns, and we would be more than happy to answer any remaining concern.**

---

> ### Author Response · Authors · 2025-08-05
>
> Dear Reviewer,
>
> Thank you for your time and review! We sincerely thank you for your time, effort, and thoughtful review and for recognizing the novelty of our framework. We hope our responses have fully addressed your concerns. We would be very happy to continue the discussion and answer any remaining questions or concerns. If you find our response satisfactory, we would be grateful if you would consider raising your score.

---

> > ### Comment · Reviewer_P97x · 2025-08-06
> >
> > Thank you for your clarifications. Kindly include these examples/illustration in the final version of the paper. I will keep my recommendation of acceptance (4).

---

### Official Review · Reviewer_aBwg · 2025-07-02

**Clarity:** 1
**Significance:** 2
**Originality:** 2
**Rating:** 4
**Confidence:** 2

**Summary:**

The paper proposes a conceptual framework that treats a model and the optimiser that trains it as a stack of nested optimisation problems. The authors formalise "prime" modules (that decompose to themselves) and a general nesting relation, as well as interpret popular optimisers as associate memory learners. They then prove a generic theorem that a neural network represented as an instance of NL can at least learn one-step gradient descent. Finally, they propose HOPE, a neural learning module that is dynamically updating its key/value/query projections using a single internal objective.

**Questions:**

- How is NL different from bi-level/meta-learning formalisms?
- What does it mean for a model to be "mathematically white-box" (see abstract)? How does this yield claimed interpretability gains?

**Ethical Concerns:**

["NO or VERY MINOR ethics concerns only"]

**Final Justification:**

The framework of jointly considering architectures and optimizers represents an intellectually appealing direction that could inspire future research. The work would benefit from a major revision addressing presentation concerns, but the core insights and empirical validation justify acceptance.

**Limitations:**

Yes

**Quality:**

3

**Strengths And Weaknesses:**

**Strenghts:**
- The conceptual framework of considering architectures and optimisers jointly is intellectually appealing
- HOPE achieves small but consistent gains over strong recurrent baselines

**Weaknesses:**
- The induction proof for theorem 1 is largely intuitive. It assumes that nesting objectives automatically provides a valid inner-gradient update without articulating convergence conditions under which this is guaranteed.
- The paper was overall hard to follow

---

> ### Author Rebuttal · Authors · 2025-07-31
>
> We thank the reviewer for their time and valuable comments. We are glad that you found our framework intellectually appealing. We respond to your comments below:
>
>
> > The induction proof for theorem 1 is largely intuitive. It assumes that nesting objectives automatically provides a valid inner-gradient update without articulating convergence conditions under which this is guaranteed.
>
> **Response:** Please note that our proof is based on induction and is by construction. In fact, we do not rely on convergence conditions and it is completely possible that the nested optimization problem does not converge to a global optima. This theorem, however, indicates that **one step of gradient descent** in the inner problem can be learnt by the outer problem.
>
>
>
> > The paper was overall hard to follow
>
> **Response**: We thank the reviewer for bringing this point to our attention. We adhere to extensively proofreading the paper and clear points that are unclear. Also, if there is a specific part that you want us to clarify or improve, we would be happy to provide additional details and improve it in the final version of the paper.
>
> Having said that, we would like to explain that our paper advocates for a completely new learning paradigm.  Therefore, we need to introduce several new terms and concepts,which might not be fully aligned with the readers’ prior viewpoint. This naturally makes the paper more difficult to follow compared to the majority of papers in the community.. We kindly ask you to consider this point in your evaluation. We will, however, follow your suggestion and will do our best to improve the writing in the final version of the paper as much as we can.
>
>
>
> > How is NL different from bi-level/meta-learning formalisms?
> **Response:** This is a great question. The main differences are as follows:
>
> - As illustrated in Figure 1, levels are not necessarily fully nested and we might have parallel building blocks at the same level. An example of this is to optimize linear recurrent models with gradient descent with momentum (we have provided this example in Sec. 2.2).  On the other hand, meta-learning always consists of an inner-loop and an outer-loop.
> - Even considering fully nested layers, NL is not necessarily a bi-level optimization problem and allows to have several nested levels. Please note that even extending meta-learning to the case of meta-meta-...-meta learning, these nested optimizations are different as here there is a flexibility to have 1 outer-loop or each level be the outer-loop of its 1 level below. This flexibility is not in meta-meta-...-meta learning problem. From this perspective, NL is closer to hierarchical games.
> - In NL, we can have modules at different levels but each of which is the most outer level of its branch, meaning that there is no other module that meta-learn them.
> - Levels such as momentum are not fully meta-learning levels while they shape a two-level nested problem, as they can be collapsed into one single level, but with caching previous states.
>
>
> > What does it mean for a model to be "mathematically white-box" (see abstract)? How does this yield claimed interpretability gains?
>
> **Response:** This is an important question. Thank you for asking. What we wanted to refer to here was: Given NL perspective, the objective and inner process of each component is clear from the mathematical point of view. In fact, each module is now translated into optimizing an objective with gradient descent, which means that based on the objective, we directly know what is optimized by each component. This perspective further is important for alignment, in which we can simply encode any objective into the internal process of the model, making it aligned with an arbitrary value of interest.
>
> After thinking more about it, we agree with the reviewer that this terminology is not very clear. Thus, in the final version of the paper, we would like to replace it with ``white-box objective’’. We would love to hear your suggestions as well regarding this terminology.
>
> ---
> ## General Note on Contributions:
>
> ---
> Finally, we would like to kindly bring some points about the contributions of our work to your attention: (1) Please note that in our submission, we not only present a new learning paradigm that can explain and unify several well-known concepts, but we also present a new architecture that provides a noticeable performance benefit over the baselines in a wide range of benchmarks. n addition,, we study and compare the in-context learning ability of different architectures in the Appendix, which alone has often been the only focus of some similar studies.  (2) While the current focus of the community in improving architectures is on scaling the parameters and data, Nested Learning presents a new dimension to enhance the model capability, which we believe is a solid contribution. This paradigm can help design more efficient architectures with fewer parameters, and inspire more research studies to build on .  (3) Due to the recent community interest in designing alternative sequence models and understanding the in-context learning ability of  different architectures, we believe that Nested Learning is a timely contribution and its publication can help the community to better design architectures capable of in-context learning and better design architecture-specific optimizers that can lead to better performance, even for current models.
>
>
> We hope that our above responses have addressed your concerns, and we would be more than happy to answer any remaining concern.

---

> > ### Comment · Reviewer_aBwg · 2025-08-05
> >
> > Thank you for the clarifications. I will increase my score to 4.

---

> ### Author Response · Authors · 2025-08-05
>
> Dear Reviewer,
>
> Thank you for your time and review! We sincerely thank you for your time, effort, and thoughtful review and for recognizing the consistent performance gain of our model over strong recurrent baselines on multiple benchmarks. We hope our responses have fully addressed your concerns. We would be very happy to continue the discussion and answer any remaining questions or concerns. If you find our response satisfactory, we would be grateful if you would consider raising your score.

---

### Official Review · Reviewer_TV1V · 2025-07-14

**Clarity:** 3
**Significance:** 3
**Originality:** 3
**Rating:** 4
**Confidence:** 1

**Summary:**

The paper proposes a new paradigm called Nested Learning (NL), which aims to unify the long-separated fields of neural architecture design and optimization algorithms in deep learning. The authors argue that this separation is unnatural and demonstrate how a complete learning system can be viewed as a set of nested optimization problems. This framework considers meta-learning as a two-level special case and can be extended to more levels and more complex structures.

The HOPE dynamizes the traditional key and value projection matrices, allowing them to adapt at each step of sequence processing, thereby achieving in-context learning at all levels. Experimental results show strong performance on language modeling, commonsense reasoning, and long-context benchmarks.

**Questions:**

Have the authors considered a "relaxed" nested version that allows a small amount of gradient information to flow between levels?

**Ethical Concerns:**

["NO or VERY MINOR ethics concerns only"]

**Final Justification:**

The authors have sufficiently clarified my concerns regarding the gap between theory and practice (block-wise training) as a necessary trade-off. My score remains unchanged.

**Limitations:**

Yes.

**Paper Formatting Concerns:**

No.

**Quality:**

3

**Strengths And Weaknesses:**

Strengths:
The HOPE model performs exceptionally well on multiple benchmarks.

Weaknesses:
To achieve efficient training, the HOPE model appears to use a block-wise hybrid training method, which is inconsistent with its theoretical description.

---

> ### Author Rebuttal · Authors · 2025-07-31
>
> We thank the reviewer for their time and valuable comments. We are glad that you found the performance of our model exceptionally well on different benchmarks. We respond to your comments below:
>
>
> > To achieve efficient training, the HOPE model appears to use a block-wise hybrid training method, which is inconsistent with its theoretical description.
>
> **Response:** To answer your question, let us first clarify two points: (1) Similar to other types of architectures, there is a trade-off between efficiency and performance of the model. This trade-off is controlled by the size of the chunks. If we set the chunk size equal to 1, then the model becomes fully recurrent and so matches all the motivations, but this comes with the cost of hard parallelizability.  (2) Choosing larger chunk size can be viewed an approximation of the exact recurrence. In this viewpoint, as you mentioned, our theoretical description is not fully aligned with our design. However, the model still performs gradient descent to optimize the internal objective, but the updating the memory weights are happening every C tokens (which can be interpreted as splitting the input sequence, not necessarily changing the recurrence). So from this perspective, there is no gap between our theory and our design.
>
>
> > Have the authors considered a "relaxed" nested version that allows a small amount of gradient information to flow between levels?
>
> **Response**: Thank you for bringing up this setting and we agree this is indeed interesting to consider. Please note that the general formulation for Nested Learning allows such cases, however, since each box corresponds to a specific gradient flow, if there is a full gradient flow between blocks, then those two are sharing a gradient flow and so we would have considered them as a single block from the beginning. However, as you mentioned, we might want to transfer some gradient steps to other levels in a certain special design. This is an interesting direction for future work, which is out of the scope of our current work.
>
> $ ~ $
>
>
> ---
> ## General Note on Contributions:
>
> ---
> Finally, we would like to kindly bring some points about the contributions of our work to your attention: (1) Please note that in our submission, we not only present a new learning paradigm that can explain and unify several well-known concepts, but we also present a new architecture that provides a noticeable performance benefit over the baselines in a wide range of benchmarks. n addition,, we study and compare the in-context learning ability of different architectures in the Appendix, which alone has often been the only focus of some similar studies.  (2) While the current focus of the community in improving architectures is on scaling the parameters and data, Nested Learning presents a new dimension to enhance the model capability, which we believe is a solid contribution. This paradigm can help design more efficient architectures with fewer parameters, and inspire more research studies to build on .  (3) Due to the recent community interest in designing alternative sequence models and understanding the in-context learning ability of  different architectures, we believe that Nested Learning is a timely contribution and its publication can help the community to better design architectures capable of in-context learning and better design architecture-specific optimizers that can lead to better performance, even for current models.
>
>
> We hope that our above responses have addressed your concerns, and we would be more than happy to answer any remaining concern.

---

> ### Author Response · Authors · 2025-08-05
>
> Dear Reviewer,
>
> Thank you for your time and review! We sincerely thank you for your time, effort, and thoughtful review and for recognizing the `exceptionally well` performance of our model on multiple benchmarks. We hope our responses have fully addressed your concerns. We would be very happy to continue the discussion and answer any remaining questions or concerns. If you find our response satisfactory, we would be grateful if you would consider raising your score.

---

> > ### Comment · Reviewer_TV1V · 2025-08-06
> >
> > Dear Authors,
> >
> > My apologies for the delayed response. Thank you for thoroughly addressing all my previous concerns, and I will be keeping my score.
> > I strongly encourage you to release your code and setting details. This would be a great service to the community.
> >
> > Thanks again.

---

### Author Response · Authors · 2025-08-08

Dear ACs and Reviewers,

Since the discussion period is coming to an end, we would like to thank you all once again for your time and valuable feedback. We are glad that our responses helped resolve the reviewers concerns and now all the reviewers have a positive view of our work. If accepted, we adhere to using the one additional page in camera-ready version to address all the reviewers suggestions. We hope our work earns your support during the final review phase.


Best regards,
Authors

---

### Public Comment · ~A_Akhil1 · 2025-11-26
**Clarification Request on Nested Learning Objectives and Memory Levels**

Thank you for this work — the Nested Learning formulation and the HOPE architecture introduce several interesting ideas. I had a few points that I would appreciate clarification on, in order to better understand the paradigm:

1. The paper mentions that each nested memory level has its own internal objective, optimized at a different update frequency. Could you elaborate on the nature of these internal objectives? For example, are these losses explicitly defined at each level, or are they implicitly derived from the main sequence modeling objective?

2. In the discussion on block-wise (chunk-based) processing, it is stated that this is an approximation of the fully recurrent formulation. Should this chunking be viewed as an efficiency-oriented implementation detail, or does it have theoretical implications for how gradient information is separated across levels?

3. The distinction between gradient flows at different levels is a central point of the paradigm. Are there recommended constraints or principles for choosing the number of levels and their respective update rates, or is this intended to be problem-dependent?

4. Some reviewers mentioned potential clarifications on the dynamics between nested levels and the absence of backpropagation across them. Could you expand slightly on how the independence of gradient flows is ensured, especially in the presence of parameter interactions across levels?

Understanding these points would help deepen the conceptual picture of Nested Learning and how it operationally differs from meta-learning, hierarchical games, or multi-timescale recurrence.

Thank you again for your contribution, and I would be grateful for any additional insight you can provide.

---

### Decision · Program_Chairs · 2025-09-17

**Decision:**

Accept (poster)

**Comment:**

The paper proposes a novel learning approach, albeit with some similarities to meta-learning, deep declarative networks, etc., that demonstrates some promise in advancing the state-of-the-art and hence is worthy of communicating to the community. While the reviewers were on the balance positive about the paper, there were a number of issues raised that the authors have responded to, and promised to address in the final version of the paper, in particular with respect to clarity. Release of open-source software code, as promised, would also be highly valuable in communicating and understanding the method. The Area Chair agrees with the consensus of the reviewers and recommends acceptance.

---

> ### Public Comment · ~Kaixuan_WANG5 · 2026-01-19
>
> Did they promise the release of open-source code?
>
> > Release of open-source software code, as promised, would also be highly valuable in communicating and understanding the method.
>
> Didn't see any claim in the rebuttal or the paper.